# Learning Sparse Group Models Through Boolean Relaxation

**Yijie Wang**[1*]  **Yuan Zhou**[2*]  **Xiaoqing Huang**[3]  **Kun Huang**[3]  **Jie Zhang**[4]  **Jianzhu Ma**[5]

[1]Computer Science Department, Indiana University Bloomington

[2]Yau Mathematical Sciences Center and Department of Mathematical Sciences, Tsinghua University

[3]Department of Biostatistics & Health Data Science, Indiana University

[4]Department of Medical and Molecular Genetics, Indiana University

[5]Institute for AI Industry Research, Tsinghua University

`yijwang@iu.edu` `{yuan-zhou, majianzhu}@tsinghua.edu.cn`;([*] Equal contribution)

## Abstract

We introduce an efficient algorithmic framework for learning sparse group models formulated as the natural convex relaxation of a cardinality-constrained program with Boolean variables. We provide theoretical techniques to characterize the equivalent condition when the relaxation achieves the exact integral optimal solution, as well as a rounding algorithm to produce a feasible integral solution once the optimal relaxation solution is fractional. We demonstrate the power of our equivalent condition by applying it to two ensembles of random problem instances that are challenging and popularly used in literature and prove that our method achieves the exactness with overwhelming probability and the nearly optimal sample complexity. Empirically, we use synthetic datasets to demonstrate that our proposed method significantly outperforms the state-of-the-art group sparse learning models in terms of individual and group support recovery when the number of samples is small. Furthermore, we show the out-performance of our method in cancer drug response prediction.

## 1 Introduction

Sparsity is one of the most important concepts in statistical machine learning, which strongly connects to the data & computational efficiency, generalizability, and interpretability of the model. Traditional sparse estimation tasks aim at selecting sparse features at the individual level Tibshirani (1996); Negahban et al. (2012). However, in many real-world scenarios, structural properties among the individual features are assumed thanks to prior knowledge, and leveraging these structures may improve both model accuracy and learning efficiency Gramfort & Kowalski (2009); Kim & Xing (2012).

In this paper, we focus on learning the sparse group models for *intersection-closed* group sparsity, where groups of variables are either selected or discarded together. The general task of learning the sparse group models has been investigated quite a lot in literature, where most of the prior studies are based on the structured sparsity-inducing norm regularization Friedman et al. (2010); Huang et al. (2011); Zhao et al. (2009); Simon et al. (2013), which stems from Lasso Tibshirani (1996), the traditional and popular technique for a sparse estimate at the individual feature level. As reviewed in Bach et al. (2012); Jenatton et al. (2011), the structured sparsity-inducing norm is quite general and can encode structural assumptions such as trees Kim & Xing (2012); Liu & Ye (2010), contiguous groups Rapaport et al. (2008), directed-acyclic-graphs Zheng et al. (2018), and general overlapping groups Yuan et al. (2011).

Another type of approach for learning the sparse group models is to view the task as a cardinality-constrained program, where the constraint set encodes the group structures as well as restricts the number of groups of variables being selected. Baldassarre et al. Baldassarre et al. (2013) investigate the projection onto such cardinality-constrained set. However, due to the combinatorial nature of the projection, directly applying the projected gradient descent with the projection Baldassarre et al. (2013) to solve general learning problems with typical loss functions might not have good results Kyrillidis et al. (2015). Recent work Pilanci et al. (2015) studies the *Boolean relaxation* of the learning problem with cardinality constraints on the individual variables. This work Pilanci et al. (2015) can be viewed as a special case of sparse group models, where each group contains only one variable. Both the original work of Pilanci et al. (2015) and several follow-up papers Bertsimas & Parys (2020); Bertsimas et al. (2020) show that the Boolean relaxation empirically outperforms the sparse estimation methods using sparse-inducing norms (Lasso Tibshirani (1996) and elastic net Zou & Hastie (2005)), especially when the sample size is small and the feature dimension is

large. However, the results in Pilanci et al. (2015) cannot be applied to the sparse group models with arbitrary group structures.

To fill the gap, in this paper, we study the sparse group models through a cardinality-constrained program. We first propose the *Boolean relaxation for sparse group models*. We further establish an analytical and algorithmic framework for our Boolean relaxation which includes a theorem stating the equivalent condition for the relaxation to achieve the exactness (i.e., the optimal integral solution) and a rounding scheme that produces an integral solution when the optimal relaxation solution is fractional. We demonstrate the power of our equivalent condition theorem by applying it to two ensembles of random problem instances that are challenging and popularly used in literature and proving that our Boolean relaxation achieves the exactness with high probability and the nearly optimal sample complexity. Our **contributions** are threefold:

1) We propose a novel framework that uses constraints to induce *intersection-closed* group sparsity. Baldassarre et al.Baldassarre et al. (2013) investigate the projection on the group sparsity constraints. But our framework extends to any convex loss function with the group sparsity constraints.

2) We prove our framework is tight and can achieve the exactness with high probability and the nearly optimal sample complexity for two ensembles of random problem instances. This result is inspired by Pilanci et al. (2015) but our derivations and proofs are not straightforward extensions (e.g., due to the group structure, we need to analyze more complex feature-group matrices, prove new matrix concentration properties, and carefully choose different regularization parameters).

3) Empirically, we perform extensive experiments to demonstrate that our framework significantly outperforms the state-of-the-art methods when the **sample size is small** on simulated datasets. Furthermore, we show the out-performance of our framework in cancer drug response prediction.

## 1.1 RELATED WORKS

Convex programming relaxations and their rounding techniques have been widely used for approximating many combinatorial optimization problems that are computationally intractable (see, e.g., Williamson & Shmoys (2011)). The specific algorithmic technique in this work is inspired by the Boolean relaxation method introduced in Pilanci et al. (2015) for learning sparsity at the individual feature level. However, the additional group structure in our problem raises new algorithmic challenges, and both our Boolean relaxation formulation and its theoretical analysis (e.g., the equivalent condition for the exactness) are different from their counterparts Pilanci et al. (2015).

As mentioned before, sparse estimation using structured sparsity-inducing norms were thoroughly studied for learning structured sparsity under different structure assumptions motivated by various practical scenarios Friedman et al. (2010); Huang et al. (2011); Zhao et al. (2009); Simon et al. (2013); Tibshirani (1996); Bach et al. (2012); Kim & Xing (2012); Liu & Ye (2010); Rapaport et al. (2008); Zheng et al. (2018); Yuan et al. (2011); Jenatton et al. (2011). However, none of these algorithms provides the rigorous theoretical techniques as in this work to verify whether the algorithm has produced the exact optimal solution. Also, as we will show in the experiments section, our proposed method outperforms these algorithms on both synthetic and real-world datasets. In our experiments, we also compare with the elastic net method Zou & Hastie (2005), which can only control the sparsity at the individual feature level. There exist another family of structured sparsity-inducing norms Jacob et al. (2009) that aim to model the *union-closed* families of supports, where the support of the solution is a union of groups. Different from our proposed models, in which the support of the solution is the intersection of the complements of some of groups considered (*intersection-closed* group sparsity) Jenatton et al. (2009).

Another approach is to learn sparse group models by introducing the penalty functions for the constraints and applying the convex relaxation to them. Bach Bach (2010) investigate to design norms from submodular set-functions. Halabi et al. El Halabi & Cevher (2015); Halabi et al. (2018) study to induce group sparsity using tight convex relaxation of linear matrix inequalities and combinatorial penalties. Note that these works use convex regularizers to induce group sparsity while we use constraints. Halabi et al. El Halabi & Cevher (2015); Halabi et al. (2018) study general equivalent conditions to characterize the tightness of their relaxations while our theoretical results works for specific distributions where their general conditions cannot be easily verified. We use different analytical frameworks and thus the theoretical results cannot be directly compared.

## 2 BOOLEAN RELAXATION FOR SPARSE GROUP MODEL

The organization of this section is as follows. In section 2.1, we introduce the original problem and its exact boolean representation. In section 2.2, we propose the boolean relaxed program and provide the condition under which the relaxed program is guaranteed to have an integer solution and hence be tight. In section 2.3, we propose the rounding strategy if the relaxed program does not generate integral solutions.

### 2.1 SPARSE GROUP MODEL AND ITS FORMULATION VIA BOOLEAN CONSTRAINTS

We consider a learning problem for a collection of $n$ samples $\{(x_i, y_i) \in \mathbb{R}^d \times \mathcal{Y}\}_{i=1}^n$ and define the design matrix as $X \in \mathbb{R}^{n \times d}$, where $x_i^\top \in \mathbb{R}^d$ is the $i$-th row of $X$. This setup is flexible to model various problems including binary classification (where the label space $\mathcal{Y} = \{-1, +1\}$) and regression problems (where the label space $\mathcal{Y} = \mathbb{R}$). For a linear model $x \mapsto w^\top x$, our goal is to learn a sparse weight vector $w \in \mathbb{R}^d$ whose support encodes certain structures reflecting the relationships among the features which are usually defined by the prior knowledge. More formally, we need to solve the following mathematical program.

$$P^* = \min_{w \in \Theta} \left\{ F(w) := \sum_{i=1}^n f(w^\top x_i; y_i) + \frac{1}{2}\rho\|w\|_2^2 \right\}. \tag{1}$$

Here, the loss function $f(\cdot; \cdot)$ measures the prediction error by our linear model, where the common choices include the squared loss for least-squares regression, the log loss for the logistic regression, and the hinge loss for the support vector machine. The regularization term $\frac{1}{2}\rho\|w\|_2^2$ in (1) makes sure that the objective function is strongly convex and therefore has a unique optimal solution $w^* \in \mathbb{R}^d$. Finally, the constraint set $\Theta$ encodes the sparsity requirements for both individual features and groups of features. We use $g_i$ to denote the set of the indices of features in the $i$-th group and for any vector $w \in \mathbb{R}^d$, we use $w_{g_i}$ to denote the vector containing all entries of $w$ corresponding to the indices in $g_i$. We further assume that we have $b$ predefined groups and then the cardinality constraint set $\Theta$ can be written as

$$\Theta = \left\{ w \in \mathbb{R}^d \;\middle|\; \|w\|_0 \le k, \quad \sum_{j=1}^b \mathbf{1}\left[\left\|w_{g_j}\right\|_0 > 0\right] \le h \right\}, \tag{2}$$

where $\|\cdot\|_0$ denotes the $\ell_0$ norm and $\mathbf{1}[\cdot]$ denotes the indicator variable that takes the value 1 when the corresponding condition holds and 0 otherwise. The first constraint enforces the number of contributing features to be less than $k$, and the second constraint makes sure the number of groups that contain those selected features is less than $h$.

We also remark that the structured sparsity constraints defined by $\Theta$ in equation 2 is very flexible. First, the $\|w\|_0 \le k$ constraint imposes the feature-level sparsity requirement and encompasses the unstructured sparsity model (as investigated in Pilanci et al. (2015)) as a special case. Second, the group-level sparsity constraint is introduced by $\mathbf{1}\left[\left\|w_{g_j}\right\|_0 > 0\right] \le h$ covers the needs for structured sparsity arising from many practical scenarios such as neuroimaging Gramfort & Kowalski (2009); Xi et al. (2009), genomic analysis Rapaport et al. (2008); Kim & Xing (2012), and wavelet-based denoising Zhao et al. (2009); Huang et al. (2011). The groups $\{g_i\}_{i \in \{1,2,\ldots,b\}}$ can be arbitrary sets of the features and may model not only non-overlapping structured sparsity when $g_i \cap g_j = \emptyset, \forall i, j$ but also various overlapping patterns including the contiguous pattern, the block pattern, and the hierarchical pattern as reviewed in Bach et al. (2012); Jenatton et al. (2011). Note that the first term in equation 2 can be absorbed into the second term, which however will not have the sparsity control at the individual level.

Note that the structured sparse group learning problem $P^*$ defined in equation 1 involves only real variables. In the following theorem, we show that the problem can be reformulated as a convex program with additional Boolean variables and constraints, which will naturally lead to the Boolean relaxation algorithm in the later sections.

**Theorem 2.1** (Exact representation with Boolean constraints). *Suppose that for each $y \in \mathcal{Y}$, the function $t \mapsto f(t; y)$ is closed and convex. The Legendre-Fenchel conjugate of $f$ is $f^*(s; y) := \sup_{t \in \mathbb{R}}\{st - f(t : y)\}$. Then for any $\rho > 0$, the structured sparse learning problem $P^*$ in equation 1 can be represented by the following Boolean convex program*

$$P^* = \min_{(u,z) \in \Gamma} \underbrace{\max_{v \in \mathbb{R}^n} \left\{ -\frac{1}{2\rho} v^\top X D(u) X^\top v - \sum_{i=1}^n f^*(v_i; y_i) \right\}}_{H(u)}, \tag{3}$$

where $D(u) := \mathrm{diag}(u)$ is a diagonal matrix with $u \in \mathbb{R}^d$ on its diagonal, and $\Gamma$ is the constraint set for $u$ and $v$, defined as the follows,

$$\Gamma = \left\{ (u, z) \,\middle|\, \sum_{i=1}^{d} u_i \leq k, \quad \sum_{j=1}^{b} z_j \leq h, \quad u_i \leq z_j, \quad \forall i \in g_j, \quad u \in \{0,1\}^d, \quad z \in \{0,1\}^b \right\}.$$

The proof of Theorem 2.1 can be found in the supplementary materials Section A. In the theorem statement, $u$ is a vector of the Boolean indicators for the supports of the individual features and $z$ is also a vector of the Boolean indicators for the supports of the group features. $H(u)$ in equation 3 is convex in $u$ because it is the maximum of a family of functions that are linear with $u$. However, the whole program is still computationally difficult due to the Boolean constraints $u \in \{0,1\}^d$ and $z \in \{0,1\}^b$. In the next subsection, we will relax these Boolean constraints which leads to a convex program that can be efficiently solved for many popular loss functions $f$.

## 2.2 Convex Program Through Boolean Relaxation and Theoretical Conditions for Exactness

We apply interval relaxation to both Boolean vector variables $u$ and $z$, and obtain the Boolean relaxation for $P^*$ as follows.

$$P_{\mathrm{BR}} = \min_{(u,z) \in \Omega} \max_{v \in \mathbb{R}^n} \left\{ -\frac{1}{2\rho} v^\top X D(u) X^\top v - \sum_{i=1}^{n} f^*(v_i; y_i) \right\}, \tag{4}$$

where

$$\Omega = \left\{ (u, z) \,\middle|\, \sum_{i=1}^{d} u_i \leq k, \quad \sum_{j=1}^{b} z_j \leq h, \quad u_i \leq z_j, \quad \forall i \in g_j, \quad u \in [0,1]^d, \quad z \in [0,1]^b \right\}.$$

$P_{\mathrm{BR}}$ is a convex program and can be solved by the sub-gradient-based optimization algorithm Nesterov (2009) if the inner maximization problem can be solved efficiently. In general, $P_{\mathrm{BR}}$ can also be converted into a minimax optimization problem and solved by methods in Lin et al. (2020).

We now investigate when $P_{\mathrm{BR}}$ achieves the exact solution of $P^*$, under the assumption that the groups are non-overlapping. Note that $P_{\mathrm{BR}}$ is a relaxation of $P^*$ as defined in equation 3. The relaxation is exact if and only if the optimal solution in $P_{\mathrm{BR}}$ also happens to be integral and therefore feasible in $P^*$. The following theorem (proved in the supplementary materials Section B.1) characterizes the equivalent condition for the exactness.

**Theorem 2.2.** *Suppose that each feature belongs to only one group and the optimal integral solution $(\hat{u}, \hat{z})$ for $P^*$ as defined in equation 3 selects $k$ features and $h$ groups, then the optimal solution of $P_{\mathrm{BR}}$ also recovers $(\hat{u}, \hat{z})$ if and only if there exists non-negative $\lambda_k$ and $\lambda_h$, such that*

$$\hat{v} \in \arg\max_{v \in \mathbb{R}^n} \left\{ -\frac{1}{2\rho} v^\top X D(\hat{u}) X^\top v - \sum_{i=1}^{n} f^*(v_i; y_i) \right\} \tag{5}$$

1. *For each group $i$ such that $\hat{z}_i = 1$, it holds that*
$$\forall p \in g_i, \hat{u}_p = 1 \Rightarrow (X_p^\top \hat{v})^2 > \lambda_k \quad and \tag{6}$$
$$\forall p \in g_i, \hat{u}_p = 0 \Rightarrow (X_p^\top \hat{v})^2 \leq \lambda_k. \tag{7}$$

2. *For each group $i$ such that $\hat{z}_i = 1$, it holds that*
$$\sum_{p \in g_i, \hat{u}_p = 1} ((X_p^\top \hat{v})^2 - \lambda_k) > \lambda_h. \tag{8}$$

3. *For each group $i$ such that $\hat{z}_i = 0$,*
$$\sum_{p \in g_i} \max\{(X_p^\top \hat{v})^2 - \lambda_k, 0\} \leq \lambda_h. \tag{9}$$

*Here, $X_p$ denotes the $p$-th column of the design matrix $X$.*

**The special case of least-squares regression.** Among all candidate choices of the loss function $f$, the squared loss $f(t; y) = \frac{1}{2}(t - y)^2$ for least-squares regression is the most important and popular one with many real-world applications Kim et al. (2007); Nguyen & Rocke (2002); Boulesteix & Strimmer (2007); Fort & Lambert-Lacroix (2004), and the corresponding Legendre-Fenchel conjugate becomes $f^*(s; y) = \frac{s^2}{2} + sy$. In this special case of the structured sparse learning for least-squares

regression, the relaxed convex program $P_{\text{BR}}$ becomes the following form.

$$L_{\text{BR}} = \min_{(u,z)\in\Omega} \left\{ G(u) := y^\top \left( \frac{1}{\rho} X D(u) X^\top + I \right)^{-1} y \right\}. \tag{10}$$

The detailed derivation of equation 10 can be found in the supplementary materials Section B.2. We let $S$ denote the support of the unique optimal solution $u^*$ to the original Boolean program

$$L^* := \min_{(u,z)\in\Gamma} \{ G(u) \} \tag{11}$$

and define the $n \times n$ matrix $M$ by $\quad M := \left( I_n + \rho^{-1} X_S X_S^\top \right). \tag{12}$

Now we are ready to apply Theorem 2.2 to the squared loss function and derive the sufficient and necessary condition for the exactness of $L_{\text{BR}}$ (assuming non-overlapping groups), as follows.

**Corollary 2.3.** *Suppose that each feature belongs to only one group and the optimal integral solution $(\hat{u}, \hat{z})$ selects $k$ features and $h$ groups, then $L_{\text{BR}} = L^*$ if and only if there exist non-negative $\lambda_k$ and $\lambda_h$, such that*

*1. For each group $i$ such that $\hat{z}_i = 1$, it holds that*

$$\forall p \in g_i, \hat{u}_p = 1 \Rightarrow (X_p^\top M y)^2 > \lambda_k \quad and \tag{13}$$

$$\forall p \in g_i, \hat{u}_p = 0 \Rightarrow (X_p^\top M y)^2 \le \lambda_k. \tag{14}$$

*2. For each group $i$ such that $\hat{z}_i = 1$, it holds that*

$$\sum_{p \in g_i, \hat{u}_p=1} \left( (X_p^\top M y)^2 - \lambda_k \right) > \lambda_h. \tag{15}$$

*3. For each group $i$ such that $\hat{z}_i = 0$,*

$$\sum_{p \in g_i} \max\{ (X_p^\top M y)^2 - \lambda_k, 0 \} \le \lambda_h. \tag{16}$$

The proof of Corollary 2.3 can be found in the supplementary materials Section B.3. Corollary 2.3 creates an analysis framework for the exactness of the Boolean relaxation $L_{\text{BR}}$ where one only has to construct two scalars $\lambda_k$ and $\lambda_h$ and verify the conditions in equation 14, equation 15, and equation 16. In Section 3, we will follow this framework to theoretically prove the exactness of $L_{\text{BR}}$ for several classes of problem instances that are popularly studied in the literature, demonstrating the power of Corollary 2.3.

### 2.3 RANDOMIZED ROUNDING

When the Boolean relaxation is not exact (i.e., the optimal solution of the Boolean relaxation turns out to be fractional), we describe in this section a rounding method to recover an integral solution. We will apply randomized rounding, a state-of-the-art technique for converting fractional solutions to integer solutions with provable approximation guarantees Pilanci et al. (2015), to the solution of the relaxed problem $P_{\text{BR}}$. Given the fractional solution $\bar{u} \in [0, 1]^d$ and $\bar{z} \in [0, 1]^b$, our goal is to recover a feasible Boolean solution $u \in \{0, 1\}^d$ and $z \in \{0, 1\}^b$. For simplicity of exposition, we show the rounding scheme for the case when each feature belongs to exactly one group. However, the algorithm could be easily generalized to the cases of overlapping groups. In our rounding algorithm, we first generate a feasible Boolean solution at the group level $z \in \{0, 1\}^b$. For each group $j$, we independently set $z_j$ according to the following probability distribution:

$$\Pr[z_j = 1] = \bar{z}_j \quad \text{and} \quad \Pr[z_j = 0] = 1 - \bar{z}_j. \tag{17}$$

Once the groups are decided, the $u_i$ is set to 0 for each feature $i$ that belongs to a non-selected group. For each selected group $g_j$ and for each feature $i$ that belongs to the group, we independently set $u_i$ according to the following probability distribution:

$$\Pr[u_i = 1] = \frac{\bar{u}_i}{\bar{z}_j} \quad \text{and} \quad \Pr[u_i = 0] = 1 - \frac{\bar{u}_i}{\bar{z}_j}. \tag{18}$$

It is easy to verify that the Boolean solution generated by the method above matches the fractional solution in the sense of expectation $\mathbb{E}[z] = \bar{z}$ and $\mathbb{E}[u] = \bar{u}$, and furthermore their expected $\ell_0$ norms are given by $\mathbb{E}[\|z\|_0] = \sum_{j=1}^b \Pr[z_j = 1] = \sum_{j=1}^b \bar{z}_j \le h$, and $\mathbb{E}[\|u\|_0] = \sum_{i=1}^d (\Pr[\tilde{u}_i = 1, \tilde{z}_j = 1] + \Pr[\tilde{u}_i = 1, \tilde{z}_j = 0]) = \sum_{i=1}^d \bar{u}_i \le k$.

With these expectation bounds in hand, applying standard concentration inequalities, we can show that if we let $\mathcal{G} = \max_j |g_j|$ be the size of the largest group, for any $\delta \in (0, 1/3)$, with probability at least $(1 - \exp(\Omega(-h\delta^2)) - \exp(-\Omega(k^2\delta^2/(b\mathcal{G}^2))))$, we have that $\|z\|_0 \le (1 + \delta)h$ and $\|u\|_0 \le (1 + \delta)k$. This means that when the group sizes are relatively small, our randomized rounding scheme produces a nearly optimal solution with high probability. Finally, once we have obtained

the integral solution $u$, the weight vector $w$ for the original problem equation 1 can be computed by $w := \arg\min_{w \in \mathbb{R}^d} F(D(\tilde{u})w)$.

**Value guarantees for least-squares regression.** For least-squares loss, we are also able to establish theoretical guarantees for the value (i.e., $H(u)$ as defined in equation 3) of our rounding scheme. Without loss of generality, let us assume the columns of the design matrix $X$ are normalized, i.e., $\|X_p\|_2 \leq 1$ for $p \in \{1, 2, \ldots, d\}$ and $\|y\|_2 = 1$. We have the following theorem and its proof is in the supplementary materials Section C.

**Theorem 2.4.** *Let $(\bar{u}, \bar{z})$ be the optimal solution to the relaxed program. Let $r_z$ be the number of fractional entries in $\bar{z}$ and let $r_u$ be the number of fractional entries in $\bar{u}$. Let $(u, z)$ be the integral solution returned by our rounding scheme. For any $\delta > 0$, with probability $(1 - \delta)$, it holds that*

$$H(u) - P^* \leq O\left(\frac{1}{\rho}\left(\sqrt{r_z \log(r_z/\delta)}\mathcal{G} + \sqrt{r_u \log(r_u/\delta)}\right)\right).$$

# 3 THEORETICAL GUARANTEES OF $L_{\text{BR}}$ ON ENSEMBLES OF RANDOM INSTANCES

In this section, we apply Corollary 2.3 to prove our relaxed program is tight and can achieves the exactness with high probability and the nearly optimal sample complexity for two ensembles of random problem instances.

We focus on the case of least-squares regression and its corresponding relaxation $L_{\text{BR}}$. We will introduce two ensembles of random problem instances and theoretically analyze the performance of our $L_{\text{BR}}$ on them. The first random ensemble has been popularly used in literature to evaluate the $\ell_1$ Sparse Group Lasso algorithms Simon et al. (2013); Friedman et al. (2010). The second random ensemble is designed by us. It is more challenging compared to the first ensemble as there is more than one "optimal" weight vector $w$ at the feature level. However, the algorithm has to figure out the one with the most group sparsity. For both ensembles, we will show that our $L_{\text{BR}}$ will successfully recover both the group and feature sparsity with overwhelming probability and almost optimal sample complexity (i.e., $n$ – the number of observations).

## 3.1 RANDOM ENSEMBLE I

The first class of random instances can be generated as follows (illustrated in Fig. 1(a)). We first generate the random design matrix $X \in \mathbb{R}^{n \times d}$ with *i.i.d.* $\mathcal{N}(0, 1)$ entries. The $d$ features are divided into $b$ groups where the size of each group is $d/b$. We will construct the regression weight vector $w \in \mathbb{R}^d$ such that the first $h$ groups will have non-zero coefficients and the coefficients in the rest of the groups are 0. The number of non-zero coefficients in the $j$-th group (for $j \in \{1, 2, \ldots, h\}$) is $k_j$ and we have that $\sum_{j=1}^{h} k_j = k$, where $k$ is the total number of non-zero coefficients in $w$. For each group $j \in \{1, 2, \ldots, h\}$, we arbitrarily choose $k_j$ features in the group and randomly set the corresponding coefficient in $w$ to be $\pm\frac{1}{\sqrt{k}}$. Finally, $y = Xw + \epsilon$, where the noise vector $\epsilon \in \mathbb{R}^n$ has *i.i.d.* $\mathcal{N}(0, \gamma^2)$ entries. The goal is to identify the support of $w$ and the corresponding coefficients. The following theorem provides the theoretical guarantee that given a sufficient amount of observations, $L_{\text{BR}}$ recovers the individual and group level sparsity for this random ensemble.

**Theorem 3.1.** *Consider the random instance described above with parameters $(n, d, k, \gamma, b, h)$ and let $y = Xw + \epsilon$ be the observed response vector. Suppose that $\gamma \geq 1$. Let $\rho = n^{1/2+\delta}$ ($\delta \in (0, 1/2)$). With probability at least $(1 - d\exp(-\Omega(n^{2\delta}/(\gamma^2 k))) - d\exp(-\Omega(n^{1-2\delta}))$), the relaxed program $L_{\text{BR}}$ admits the optimal solution $u^*$ and $z^*$ where $u_i^* = \mathbf{1}[w_i \neq 0]$ and $z_j^* = \mathbf{1}[j \in \{1, 2, \ldots, h\}]$.*

We remark that the regularization parameter $\rho$ is set to $n^{1/2+\delta}$ in our theorem, while in contrast, $\rho$ was set to $\sqrt{n}$ in Pilanci et al. (2015) for the sparse learning problem without group constraints. In fact, if we are only looking for $(1 - o(1))$ success probability, we can set $\delta$ to be as large (close to $1/2$) as possible. For example, if we set $\delta = 1/2 - \frac{\log\log n}{\log n}$. The success probability is $(1 - o(1))$ as long as $n/(\gamma^2 k \log n) = \omega(1)$, which means that we only need $n = \omega((k/\gamma^2)\log(k/\gamma^2))$ to achieve the $(1 - o(1))$ success probability, which almost matches the information theoretic lower bound Wainwright (2007) up to the logarithmic factor.

## 3.2 RANDOM ENSEMBLE II

We now describe the another class of random instances which is more challenging due to the multiple candidate regression vectors (illustrated in Fig. 2(a)). We first generate a random design matrix $X \in \mathbb{R}^{n \times d}$ with two candidate $k$-sparse cross-fit regression vectors $w^{(1)}, w^{(2)} \in \mathbb{R}^d$, such that both

vectors lead to the same expected response (i.e., $Xw^{(1)} = Xw^{(2)}$). The response vector is generated by $y = Xw^{(1)} + \epsilon$ where $\epsilon \in \mathbb{R}^n$ has *i.i.d.* $\mathcal{N}(0, \gamma^2)$ entries. The $d$ feature dimensions are divided into $b$ groups of the same size $d/b$ (for simplicity we assume that $d$ is a multiple of $b$). The groups are arranged in a way so that the non-zero coefficients of $w^{(1)}$ span $h$ groups (where $h \cdot d/b \geq k$), and the non-zero coefficients of $w^{(2)}$ span $\zeta h$ groups ($\zeta > 1$). Given $(X, y)$ (after randomly permuting the indices of the coordinates and the groups), the goal is to identify the support of $w^{(1)}$ since it is sparser at the group level.

We next specifically describe how we generate $w^{(1)}, w^{(2)}$ and design the groups. For any vector $w \in \mathbb{R}^k$ with no non-zero entries, we let the first $k$ entries of $w^{(1)}$ filled by $w$ and the rest filled by 0; we also let the $(k+1)$-th to the $2k$-th entries of $w^{(2)}$ filled by $w$ and the rest filled 0. We let each of the first $h$ groups contain $k/h$ non-zero identical coordinates of $w^{(1)}$, and let each of the next $\zeta h$ groups contain $k/(\zeta h)$ non-zero identical coordinates of $w^{(2)}$. We finally fill the $b$ groups with the coordinates numbered from $(2k+1)$ to $d$ so that each group has the same size $d/b$.

We next describe how we generate the design matrix $X$. We first generate a random orthogonal matrix $Q$ such that $Qw = w$. This can be done by first fixing an arbitrary orthogonal matrix with $\frac{w}{\|w\|_2}$ as its first column (i.e., letting $P = [\frac{w}{\|w\|_2}, \beta_1, \beta_2, \ldots, \beta_{k-1}]$), generating a random $(k-1) \times (k-1)$ orthogonal matrix $Q'$, and finally letting $Q = P \cdot \mathrm{diag}(1, Q') \cdot P^\top$. We then generate the matrices $X_1 \in \mathbb{R}^{n \times k}$ and $X_3 \in \mathbb{R}^{n \times (d-2k)}$ with *i.i.d.* $\mathcal{N}(0, 1)$ entries. Let $X_2 = X_1 Q$. Since $Q$ is orthogonal, it is easy to see that in the the marginal distribution of $X_2$, each entry is also *i.i.d.* $\mathcal{N}(0, 1)$. We finally let $X = [X_1, X_2, X_3]$. One can verify that $Xw^{(1)} = X_1 w$ as well as $Xw^{(2)} = X_1 Q w = X_1 w$.

For the theoretical guarantee of $L_{\mathrm{BR}}$ on our second random ensemble, we have the following theorem.

**Theorem 3.2.** *Let $X = [X_1, X_2, X_3]$ and $y = Xw^{(1)} + \epsilon$ be a random instance described above with parameters $(n, d, k, \gamma, b, h, \zeta, w)$. Suppose there exists $\xi > 0$ such that $\xi \leq |w_i| \leq \zeta^{1/4}\xi$ for all $i \in \{1, 2, \ldots, k\}$. Also suppose that $\gamma \geq 1$. Let $\rho = n^{1/2+\delta}$ ($\delta \in (0, 1/2)$). For large enough constant $\zeta$, with probability at least $(1 - d\exp(-\Omega(n^{2\delta}\xi^2/\gamma^2)) - d\exp(-\Omega(n^{1-2\delta})))$, the relaxed program $L_{\mathrm{BR}}$ admits the optimal solution $u^*$ and $z^*$ where $u_i^* = \mathbf{1}[w_i^{(1)} \neq 0]$ and $z_g^* = \mathbf{1}[\exists i \in g : w_i^{(1)} \neq 0]$. Here, we use $g$ to denote both the index of a group and the set of the features included in the group.*

We first remark that the smallest possible value of $\zeta$ for the theorem to hold can be made arbitrarily close to 1 (but greater than 1). This relaxation would only affect the constant coefficients in the $\Omega(\cdot)$ notations in the success probability bound. Also, similarly to the remark in Section 3.1, we may set $\delta = 1/2 - \frac{\log\log n}{\log n}$. The success probability is $(1 - o(1))$ as long as $n\xi^2/(\gamma^2 \log n) = \omega(1)$. Since $\xi^2$ is usually $\Theta(1/k)$, again, we only need $n = \omega((k/\gamma^2)\log(k/\gamma^2))$ to achieve the $(1 - o(1))$ success probability, almost matching the information theoretic lower bound Wainwright (2007) up to the logarithmic factor.

## 4 EXPERIMENTS

In this section, we perform extensive experiments to investigate the effectiveness of the proposed sparse group models under the setting of $\ell_2^2$-regularized least-squares regression specified in equation 10. We use both simulated datasets (non-overlapping groups) introduced in Sections 3.1 and 3.2 and a real-world application (overlapping groups) in cancer to evaluate the performance. To access the performance on simulated data, we compute the recovery accuracy of both individual supports and group supports, which are defined as $A_I(w)$ and $A_G(w)$, respectively:

$$A_I(w) := \frac{|\{i : w_i \neq 0, w_i^{\mathrm{true}} \neq 0\}|}{|\{i : w_i^{\mathrm{true}} \neq 0\}|}, \qquad A_G(w) := \frac{|\{j : w_{g_j} \neq \mathbf{0}, w_{g_j}^{\mathrm{true}} \neq \mathbf{0}\}|}{|\{j : w_{g_j}^{\mathrm{true}} \neq \mathbf{0}\}|}. \qquad (19)$$

Here $w_i^{\mathrm{true}}$ is the $i$th element of the ground-truth vector $w^{\mathrm{true}}$ and $w_{g_j}^{\mathrm{true}}$ is the weight vector for group $g_j$. We apply the projected Quasi-Newton method Schmidt et al. (2009) to efficiently solve equation 10. The details related to the optimization could be found in supplementary materials Section E. If the solution of equation 10 is not integral, we use the rounding scheme proposed in section 2.3. For the non-overlapping setting, we compare the performance of our method with Sparse Group Lasso (SGL) Simon et al. (2013), Sparse G-group cover (SGCover) El Halabi & Cevher (2015), and $\mathrm{SGL}_\infty$ (group level sparsity using $\ell_\infty$ norm and individual level sparsity using $\ell_1$ norm); for the overlapping group setting, we compare with (SGL-Overlap) Yuan et al. (2011) and SGCover El Halabi & Cevher (2015). We also compare our method with Elastic Net (ENet) Zou &

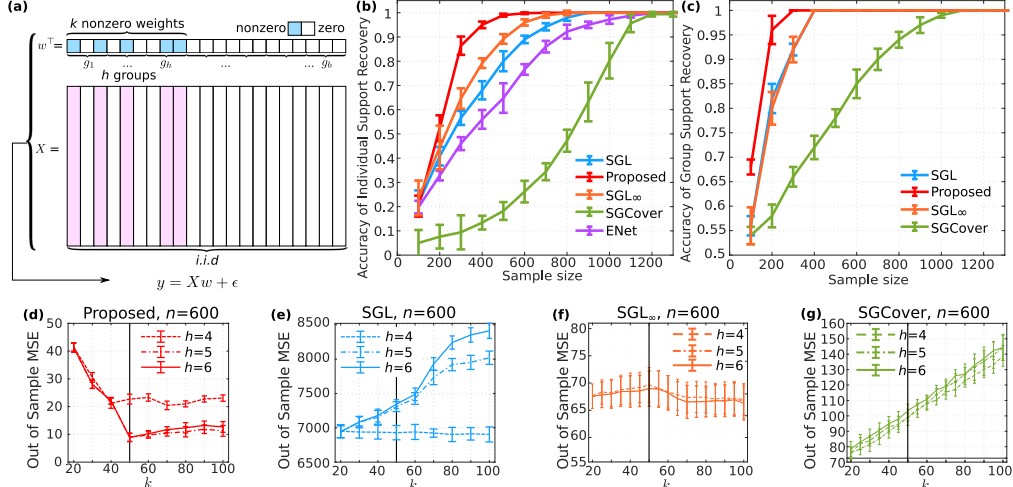

Figure 1: (a) Illustration of the data generation process for Random Ensemble I. (b) $A_I$ as $n$ increases. (c) $A_G$ as $n$ increases. We average results over 100 datasets and the error bar means 95% confidence interval. (d)–(g) Estimate $k$ and $h$ by out-of-sample of MSE by different methods. The black vertical line indicates the true $k$. The results are averaged over 10 datasets and the error bar means standard deviation.

Hastie (2005) which is the state-of-the-art method for detecting sparse features at the individual level. All experiments run on a computer with 8 cores 3.7GHz Intel CPU and 32 GB RAM.

## 4.1 SIMULATION EVALUATION OF RANDOM ENSEMBLE I

We first consider the simulation setting described in Section 3.1 in which we use $d = 1000$, $b = 10$, $k = 50$ ($k_i = 10, \forall i \in \{1, 2, \ldots, 5\}$), $h = 5$, and $\gamma = 2.5$ to generate the simulation data whose signal-noise-ratio (SNR) is around 4. We evaluate the performance of all the methods on recovering the ground truth weight vector $w^{\mathrm{true}}$ with $k = 50$ contributing features distributed in $h = 5$ groups. **Feature selection with given support sizes $k$ and $h$:** We first consider the case when $k$ and $h$ are given and equal to the ground-truth for all methods, while all other hyper-parameters are selected by cross-validation. It is hard to let SGL, SGCover, and $\mathrm{SGL}_\infty$ to select exact $k$ individual features and $h$ groups of features so we indirectly control $k$ and $h$ by sweeping the regularization parameters. For cases where these methods do not yield exact $k$ individual features and $h$ groups of features, we rank their results and get the top $k$ individual features and $h$ groups of features instead. The same procedure was adopted to ENet to select $k$ features. In this setting, we do not need to worry about false discovery rate (FDR), because it is complementary to the accuracy defined in equation 19. As shown in Fig. 1(b) and (c), $A_I$ and $A_G$ of all competing methods converge to 1 with the increasing sample size $n$. However, our proposed method converges the fastest among all the methods, indicating its effectiveness on recovering the structured sparsity when the sample size is small. We further conduct similar experiments with larger $\gamma$ and show the results in the supplementary materials Section G. Based on these results, we could confirm that our proposed method outperforms conventional methods in selecting contributing features at both the individual level and group level.

**Estimation of support sizes $k$ and $h$:** In the real-world scenario, we might not know the number of contributing features and groups. Typically, they could be selected by cross-validation based on the out-of-sample mean square error (MSE). Motivated by this practical need, in this study, we also investigate whether the competing methods could accurately recover the ground truth $k$ and $h$ based on the out-of-sample of MSE. As shown in Fig. 1(d)–(g), only our proposed method is able to provide the smallest MSE at $h = 5$. In addition, for the number of contributing feature $k$, only our method achieves the smallest MSE around the number of features $k = 50$.

**Interpreting the superior performance of our method.** All the competing methods are norm-based and they rely on the regularization factor before the penalty norm to adjust to the support size (group number and feature number) requirements. However, this connection is not explicit as our Boolean relaxation where we directly require that $\sum_{i=1}^d u_i \leq k$ and $\sum_{i=1}^b z_i \leq h$. Together with our randomized rounding scheme, our Boolean relaxation method might be at an advantageous position to utilize the prior knowledge on support sizes and/or recover the support sizes from the out-of-sample MSE.

## 4.2 SIMULATION EVALUATION OF RANDOM ENSEMBLE II

Next, we consider a more challenging simulated data generated from the procedure introduced in Section 3.2. We set $d = 500$, $k = 80$, $h = 10$, $\zeta = 4$, and $\gamma = 0.1$ as described in Section 3.2. As shown in Fig. 2(a), both $w^{(1)}$ and $w^{(2)}$ are "optimal" for the linear regression problem. However, the

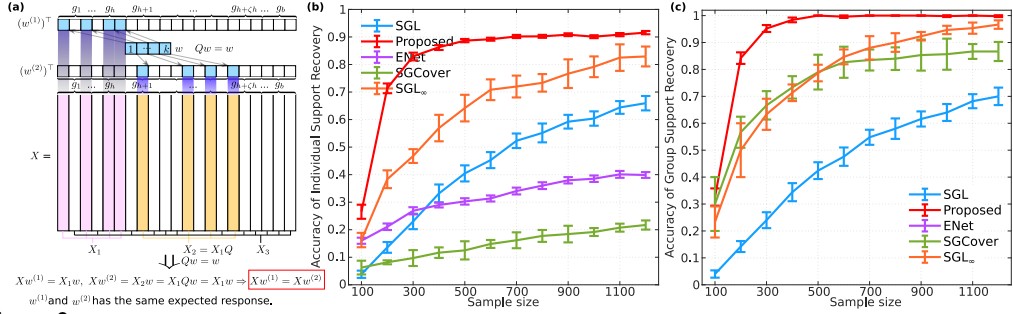

**Figure 2:** (a) Illustration of the data generation process for Random Ensemble II. (b) $A_I$ as $n$ increases. (c) $A_G$ as $n$ increases. We average results over 100 datasets and the error bar means 95% confidence interval.

supports of $w^{(1)}$ are in $h = 10$ groups but the supports of $w^{(2)}$ are in $\zeta h = 40$ groups. The goal is to test whether each method can recover the solution $w^{(1)}$, which is more sparse on the group level. We control the parameters of each method to make it select $k = 80$ features in $h = 10$ groups. As shown in Fig. 2(b) and (c), the recovery accuracy ($A_G$ and $A_I$) of the proposed method rapidly converges to 1 and 0.93 when more training samples are provided. Surprisingly, the recovery accuracy of all the rest of the competing methods converge very slowly. Note that here we do not need to consider false discovery rate (FDR) because each method selects exact $k$ features in $h$ groups, therefore, FDR is complementary to the accuracy. Overall, under the setting of Random Ensemble II, the performance improvement of our method over the-state-of-art algorithms is significant in terms of both recovery accuracy.

### 4.3 Cancer Drug Response Prediction

We further adapt our model and algorithm on a real-world application to predict the drug response. The goal of the task is to find the essential genes and pathways that are responsible for the ineffectiveness of cancer therapy. For this task, we chose $\ell_2^2$-regularized least-square regression model to predict the continuous value of drug response.

We collect drug response data from the Cancer Therapeutics Response Portal (CTRP) v2 and the Genomics of Drug Sensitivity in Cancer (GDSC) database Seashore-Ludlow et al. (2015); Yang et al. (2013) with 684 chemotherapy

**Table 1:** Result comparison for IMATNIB.

| Method | $k$(s.d.) | $h$(s.d.) | Out-of-sample MSE $\pm 95\%$CI |
|---|---|---|---|
| Proposed | 46 | 7 | 32.6$\pm$ 2.2 |
| SGL-Overlap | 92 (5.4) | 19 (0.5) | 46.9$\pm$3.7 |
| ENet | 60 (8.2) | 18 (2.3) | 39.6 $\pm$ 4.2 |
| SGCover | 321 (10.5) | 13 (1.7) | 55.4 $\pm$ 6.9 |

and targeted therapy drugs. For each drug, we create a separate machine learning task to predict the drug response from the expression value of each gene for different tumor samples. In total, we include 1,225 tumor samples and use the gene expression value of 2,369 genes. We focus on the signal transduction pathways which are mined and collected by the Reactome database Jassal et al. (2020). We only consider pathways that contain more than 10 genes and less than 80 genes. We collect 207 pathways (gene groups), in which the average number of genes in each group is 28.9. The gene expression data are extracted from CCLE Barretina et al. (2012). For each drug (machine learning task), we hold 20% of the samples as the test set and used the remaining samples as training and validation set. For each competing method, we use standard cross-validation to determine the hyper-parameter based on the out-of-sample square error MSE on the validation set. Here we only show the performance of the drug IMATNIB as a representative and put the performance of other drugs in the supplementary materials Section I. Table 1 illustrates the estimation of $k$ and $h$ and the out-of-sample MSE on the test set for drug IMATNIB with 10 bootstrap samples. We do not compare with SGL$_\infty$ because the SpaSM package Sjöstrand et al. (2018) we used to solve SGL$_\infty$ cannot handle overlapping groups. We find that our proposed method achieves the smallest out-of-sample MSE as well as selects the smallest number of genes and pathways. More importantly, as shown in supplementary materials Table S3, the selected pathways are all well associated with the drug response and functional mechanisms of IMATNIB in different types of tumor cells supported by rich literature. We also make predictions for other drugs and show the results in Table. S4.

### 5 Conclusion

In this paper, we propose a novel convex framework for learning structured sparsity. We provide theoretical tools to verify the exactness of the solution of the relaxation, and a rounding algorithm to produce the feasible integral solution when the relaxation solution is fractional. For the case of least-squares loss, we perform extensive experiments to demonstrate the effectiveness of the proposed framework.

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

# Supplementary Materials

## A   PROOF OF THEOREM 2.1

*Proof Theorem 2.1.* Let $u_i \in \{0, 1\}$ indicate whether the $i$-th feature is selected and $u_{g_j}$ be a vector containing $u_i, i \in g_j$. We then define $D(u) := \text{diag}(u)$ and $D(u_{g_j}) := \text{diag}(u_{g_j})$. Considering the change of variable $w = D(u)w$, we find the original problem equation 1 is equivalent to

$$P^* = \min_{\substack{\|D(u)w\|_0 \le k \\ \sum_{j=1}^b \mathbf{1}\left[\left\|D(u_{g_j})w_{g_j}\right\|_0 > 0\right] \le h}} \left\{ \sum_{i=1}^n f(w^\top D(u)x_i; y_i) + \frac{1}{2}\rho\|D(u)w\|_2^2 \right\}. \tag{20}$$

We further introduce $z_j \in \{0, 1\}$ to indicate whether the group of features $g_j$ is selected and obtain the following equivalent formulation

$$P^* = \min_{\substack{\|D(u)w\|_0 \le k \\ \left\|D(u_{g_j})w_{g_j}\right\|_0 \le z_j, \forall j \\ \sum_{j=1}^b z_j \le h}} \left\{ \sum_{i=1}^n f(w^\top D(u)x_i; y_i) + \frac{1}{2}\rho\|D(u)w\|_2^2 \right\}. \tag{21}$$

We further split $w$ and $u$ and then equation 21 becomes

$$P^* = \min_{(u,z)\in\Gamma} \min_{w\in\mathbb{R}^d} \left\{ \sum_{i=1}^n f(w^\top D(u)x_i; y_i) + \frac{1}{2}\rho\|w\|_2^2 \right\}, \tag{22}$$

where

$$\Gamma = \left\{ (u, z) \middle| \sum_{i=1}^d u_i \le k, \quad \sum_{j=1}^b z_j \le h, \quad u_i \le z_j, \forall i \in g_j, \quad u \in \{0, 1\}^d, \quad z \in \{0, 1\}^b \right\}.$$

It is easy to verify that equation 22 achieves the same objective function value of equation 1 at the same unique optimal solution $w^*$. It remains to prove the inner minimization is equivalent to

$$\min_{w\in\mathbb{R}^d} \left\{ \sum_{i=1}^n f(w^\top D(u)x_i; y_i) + \frac{1}{2}\rho\|w\|_2^2 \right\} = \max_{v\in\mathbb{R}^n} \left\{ -\frac{1}{2\rho}v^\top X D(u) X^\top v - \sum_{i=1}^n f^*(v_i; y_i) \right\}. \tag{23}$$

Replacing $f$ by its Legendre-Fenchel conjugate $f^*$, we have

$$\min_{w\in\mathbb{R}^d} \max_{v\in\mathbb{R}^n} \left\{ \sum_{i=1}^n w^\top D(u)x_i \cdot v_i - f^*(v_i : y_i) + \frac{1}{2}\rho\|w\|_2^2 \right\}. \tag{24}$$

Under the stated assumptions, strong duality must hold and therefore minimum and maximum can be exchanged.

$$\max_{v\in\mathbb{R}^n} \min_{w\in\mathbb{R}^d} \left\{ \sum_{i=1}^n w^\top D(u)x_i \cdot v_i - f^*(v_i : y_i) + \frac{1}{2}\rho\|w\|_2^2 \right\}. \tag{25}$$

The objective function is strongly convex with respect to $w$. Hence, we can obtain the unique minimizer $w^* = \frac{1}{\rho}\sum_{i=1}^n D(u)x_i v_i$. Substituting $w^*$ yields equation 23. $\qquad\square$

## B   PROOFS AND DEVIATIONS FOR THE CONVEX FORMULATION

### B.1   PROOF OF THEOREM 2.2

To prove Theorem 2.2, let us first provide the sufficient and necessary conditions for $P_{\text{BR}}$ to have integral solutions in the following Lemma.

**Lemma B.1.** *Suppose that each feature belongs to exactly one group. Then suppose that the integral solution $(\hat{u}, \hat{z})$ of equation 3 selects exactly $k$ features and $h$ groups. This solution is also the optimal solution of the relaxed program $P_{\text{BR}}$ if and only if there exist non-negative $\{\lambda_{u_p\le1}, \lambda_{u_p\ge0}\}_{p\in[d]}$, $\{\lambda_{z_i\le1}, \lambda_{z_i\ge0}\}_{i\in[b]}$, $\{\lambda_{u_p\le z_i}\}_{\forall p\in g_i}$, $\lambda_k$, and $\lambda_h$, such that*

$$\hat{v} \in \arg\max_{v\in\mathbb{R}^n} \left\{ -\frac{1}{2\rho}v^\top X D(\hat{u}) X^\top v - \sum_{i=1}^n f^*(v_i; y_i) \right\} \tag{26}$$

$$\lambda_{u_p \leq 1} - \lambda_{u_p \geq 0} + \lambda_k = (X_p^\top \hat{v})^2 - \lambda_{u_p \leq z_i}, \qquad \forall p \in g_i; \qquad (27)$$

$$\lambda_{z_i \leq 1} - \lambda_{z_i \geq 0} + \lambda_h = \sum_{p \in g_i} \lambda_{u_p \leq z_i}, \qquad \forall i \in [b]; \qquad (28)$$

$$\lambda_{u_p \leq 1} = 0, \qquad \forall p : \hat{u}_p = 0; \qquad (29)$$

$$\lambda_{u_p \geq 0} = 0, \qquad \forall p : \hat{u}_p = 1; \qquad (30)$$

$$\lambda_{z_i \leq 1} = 0, \qquad \forall i : \hat{z}_i = 0; \qquad (31)$$

$$\lambda_{z_i \geq 0} = 0, \qquad \forall i : \hat{z}_i = 1; \qquad (32)$$

$$\lambda_{u_p \leq z_i} = 0, \qquad \forall p \in g_i : \hat{u}_p < \hat{z}_i. \qquad (33)$$

To prove Lemma B.1, we need to use the following two theorems.

**Theorem B.2** (Davis (2020)). *Suppose $\bar{x}$ is a local minimizer of $f : \mathbb{R}^d \to \mathbb{R}$ on a closed convex set $\mathcal{X} \subseteq \mathbb{R}^d$. If $f$ is differentiable at $\bar{x}$, it holds that*
$$- \nabla f(\bar{x}) \in \mathcal{N}_{\mathcal{X}}(\bar{x}). \qquad (34)$$

**Theorem B.3** (Davis (2020)). *Let $A \in \mathbb{R}^{m \times n}$ and let $\beta \in \mathbb{R}^m$. Consider the polyhedron $Q(A, \beta) = \{x | Ax \leq \beta\}$. Suppose $x \in Q(A, \beta)$, then the normal cone at $x$ is $N_{Q(A,\beta)}(x) = \{A^\top y | y \in \mathbb{R}^m$ such that $y \geq 0$ and $y^\top(\beta - Ax) = 0\}$.*

*Proof of Lemma B.1.* $P_{\text{BR}}$ is

$$P_{\text{BR}} = \min_{(u,z) \in \Omega} \underbrace{\max_{v \in \mathbb{R}^n} \left\{ -\frac{1}{2\rho} v^\top X D(u) X^\top v - \sum_{i=1}^{n} f^*(v_i; y_i) \right\}}_{F(u,z)}, \qquad (35)$$

where $\Omega = \{(u, z) | \sum_j u_j \leq k; \sum_i z_i \leq h; u_i \leq z_j; \forall i \in g_j; u \in [0,1]^d; z \in [0,1]^b\}$ is the feasible set for $(u, z)$. By Theorem B.2, we know $(\hat{u}, \hat{z})$ is optimal if and only if the following inclusion holds:

$$- \nabla F(\hat{u}, \hat{z}) \in \mathcal{N}_{\Omega}(\hat{u}, \hat{z}), \qquad (36)$$

where $\mathcal{N}_{\Omega}(\hat{u}, \hat{z})$ is the normal cone at $(\hat{u}, \hat{z})$.
Regarding the Left-Hand-Side of equation 36, by standard calculation, we have that $\partial_{u_i} F(\hat{u}) = -(X_i^\top \hat{v})^2$ ($\hat{v}$ is defined in equation 26) and $\partial_{z_i} F(\hat{z}) = 0$. Therefore,

$$- \nabla F(\hat{u}, \hat{z}) = \begin{bmatrix} (X_1^\top \hat{v})^2 \\ (X_2^\top \hat{v})^2 \\ \vdots \\ (X_d^\top \hat{v})^2 \\ 0 \\ 0 \\ \vdots \\ 0 \end{bmatrix}. \qquad (37)$$

Regarding the Right-Hand-Side of equation 36, we obtain the normal cone $\mathcal{N}_{\Omega}(\hat{u}, \hat{z})$ using Theorem B.3. Specifically, the feasible set $\Omega$ is a polyhedron that can be presented by $\Omega(A, b) = \left\{ \begin{bmatrix} \hat{u} \\ \hat{z} \end{bmatrix} \bigg| A \begin{bmatrix} \hat{u} \\ \hat{z} \end{bmatrix} \leq \beta \right\}$, where the $A$ matrix and $\beta$ vector are constructed as follows.

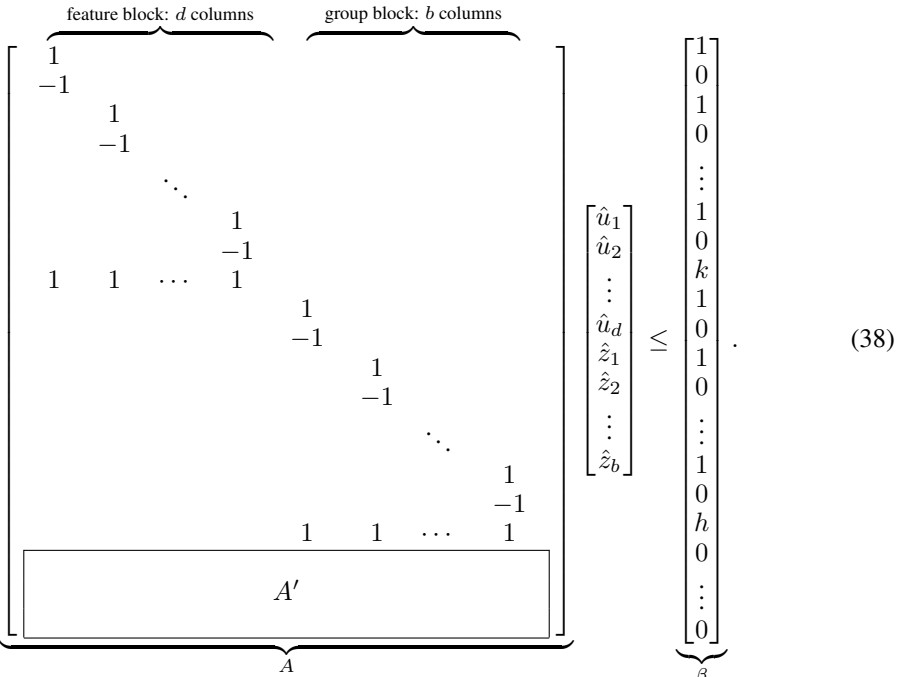

$$(38)$$

Here, $A'$ is the feature-group relation matrix where for each feature group relation $p \in g_i$, there is a row in $A'$ where the $p$-th entry in the feature block is 1, the $i$-th entry in the group block is $-1$, and all the rest entries are 0. According to Theorem B.3, we know the normal cone $\mathcal{N}_\Omega(\hat{u}, \hat{z}) = \mathcal{N}_{\Omega(A,b)}(\hat{u}, \hat{z}) = \left\{ A^\top \lambda \middle| \lambda \in (\mathbb{R}^{\geq 0})^c : \lambda^\top \left( b - A \begin{bmatrix} \hat{u} \\ \hat{z} \end{bmatrix} \right) = 0 \right\}$, where $c$ denotes the number of constraints (i.e., the row dimension of $A$). We identify the entries of $\lambda$ as follows: $\lambda_{u_p \leq 1}$ denotes the dual parameter corresponding to the constraint $u_p \leq 1$ for each feature $p$; $\lambda_{u_p \geq 0}$ denotes the dual parameter corresponding to the constraint $u_p \geq 0$ for each feature $p$; $\lambda_k$ denotes the dual parameter corresponding to the constraint $\sum_j u_j \leq k$; $\lambda_{z_i \leq 1}$ denotes the dual parameter corresponding to the constraint $z_i \leq 1$ for each group $i$; $\lambda_{z_i \geq 0}$ denotes the dual parameter corresponding to the constraint $z_i \geq 0$ for each group $i$; $\lambda_h$ denotes the dual parameter corresponding to the constraint $\sum_i z_i \leq h$; $\lambda_{u_p \leq z_i}$ denotes the dual parameter corresponding to the constraint $u_p \leq z_i$ for each feature $p$ and group $g_i$ such that $p \in g_i$.

Finally, we conclude that the equivalent condition of equation 36 is there exists $\lambda \in (\mathbb{R}^{\geq 0})^c$ such that $-\nabla F(\hat{u}, \hat{z}) = A^\top \lambda$ and $\lambda^\top \left( b - A \begin{bmatrix} \hat{u} \\ \hat{z} \end{bmatrix} \right) = 0$. By equation 37, we obtain equation 27 and equation 28 as the equivalent condition of $-\nabla F(\hat{u}, \hat{z}) = A^\top \lambda$. We also obtain equation 29, equation 30, equation 31, equation 32, and equation 33 as the equivalent condition of $\lambda^\top \left( b - A \begin{bmatrix} \hat{u} \\ \hat{z} \end{bmatrix} \right) = 0$. $\qquad\square$

With Lemma B.1 in hand, we can prove Theorem 2.2 in the following.

*Proof of Theorem 2.2.* We first prove the sufficient condition. Given $\lambda_k$ and $\lambda_h$, we only need to construct non-negative $\{\lambda_{u_p \leq 1}, \lambda_{u_p \geq 0}\}_{p \in [d]}$, $\{\lambda_{z_i \leq 1}, \lambda_{z_i \geq 0}\}_{i \in [b]}$, $\{\lambda_{u_p \leq z_i}\}_{\forall p \in g_i}$ to satisfy equation 27-equation 33. By Lemma B.1, this will establish the optimality of $(\hat{u}, \hat{z})$ in the relaxed program $P_{\text{BR}}$.

We first construct $\{\lambda_{u_p \leq 1}, \lambda_{u_p \geq 0}\}_{p \in [d]}$ and $\{\lambda_{u_p \leq z_i}\}_{\forall p \in g_i}$ as follows.

1. For each group $i$ such that $\hat{z}_i = 1$, and for each $p \in g_i$ and $\hat{u}_p = 1$, set $\lambda_{u_p \leq 1} = \lambda_{u_p \geq 0} = 0$ and $\lambda_{u_p \leq z_i} = (X_p^\top \hat{v})^2 - \lambda_k$. For each $p \in g_i$ and $\hat{u}_p = 0$, set $\lambda_{u_p \geq 0} = \lambda_k - (X_p^\top \hat{v})^2$ and $\lambda_{u_p \leq 1} = \lambda_{u_p \leq z_i} = 0$. By equation 14, one may verify that all constructed values in this step are non-negative.

2. For each group $i$ such that $\hat{z}_i = 1$, set $\lambda_{z_i \geq 0} = 0$ and $\lambda_{z_i \leq 1} = \sum_{p \in g_i, \hat{u}_p = 1}((X_p^\top \hat{v})^2 - \lambda_k) - \lambda_h$, which is non-negative due to equation 15.

3. For each group $i$ such that $\hat{z}_i = 0$, and for each $p \in g_i$ and $(X_p^\top \hat{v})^2 > \lambda_k$, set $\lambda_{u_p \leq 1} = \lambda_{u_p \geq 0} = 0$ and $\lambda_{u_p \leq z_i} = (X_p^\top \hat{v})^2 - \lambda_k$. For each $p \in g_i$ and $(X_p^\top \hat{v})^2 \leq \lambda_k$, set $\lambda_{u_p \geq 0} = \lambda_k - (X_p^\top \hat{v})^2$ and $\lambda_{u_p \leq 1} = \lambda_{u_p \leq z_i} = 0$. Observe that we always have $\lambda_{u_p \leq z_i} = \max\{(X_p^\top \hat{v})^2 - \lambda_k, 0\}$ for each $p \in g_i$.

4. For each group $i$ such that $\hat{z}_i = 0$, set $\lambda_{z_i \leq 1} = 0$ and $\lambda_{z_i \geq 0} = \lambda_h - \sum_{p \in g_i} \max\{(X_p^\top \hat{v})^2 - \lambda_k, 0\}$, which is non-negative due to equation 16.

Finally, it is straightforward to verify that equation 27-equation 33 are satisfied by our constructed $\lambda$, and therefore, by Lemma B.1, we conclude that $(\hat{u}, \hat{z})$ is the optimal solution of the relaxed program $P_{\mathrm{BR}}$.

We next prove the necessary condition. Given $P_{\mathrm{BR}}$ and $P^*$ have the same integral solution, we only need to show there exist $\lambda_k$ and $\lambda_h$ that satisfy equation 14–equation 16. By Lemma B.1 and $\hat{u}$ is the integral solution, we have

1. For each group $i$ such that $\hat{z}_i = 1$ and $\forall p \in g_i, \hat{u}_p = 1$, we have
$$(X_p^\top \hat{v})^2 = \lambda_k + \lambda_{u_p \leq 1} + \lambda_{u_p \leq z_i} \Rightarrow (X_p^\top \hat{v})^2 > \lambda_k. \tag{39}$$
For each group $i$ such that $\hat{z}_i = 1$ and $\forall p \in g_i, \hat{u}_p = 0$, we have
$$(X_p^\top \hat{v})^2 = \lambda_k - \lambda_{u_p \geq 0} \Rightarrow (X_p^\top \hat{v})^2 \leq \lambda_k. \tag{40}$$

2. For each group $i$ such that $\hat{z}_i = 1$ and all $p \in g_i, \hat{u}_p = 1$, we apply equation 39 and equation 28 and have
$$\sum_{p \in g_i, \hat{u}_p = 1}\left((X_a^\top \hat{v})^2 - \lambda_k\right) = \lambda_h + \lambda_{z_i \leq 1} \Rightarrow \sum_{p \in g_i, \hat{u}_p = 1}\left((X_a^\top \hat{v})^2 - \lambda_k\right) > \lambda_h. \tag{41}$$

3. For each group $i$ such that $\hat{z}_i = 0$, we apply equation 40 and equation 28 and have
$$\sum_{p \in g_i}\left((X_p^\top \hat{v})^2 - \lambda_k\right) = \lambda_h - \lambda_{z_i \geq 0} - \sum_{p \in g_i} \lambda_{u_p \geq 0}$$
$$\Rightarrow \sum_{p \in g_i} \max\{(X_p^\top \hat{v})^2 - \lambda_k, 0\} \leq \lambda_h - \lambda_{z_i \geq 0} - \sum_{p \in g_i} \lambda_{u_p \geq 0}$$
$$\Rightarrow \sum_{p \in g_i} \max\{(X_p^\top \hat{v})^2 - \lambda_k, 0\} \leq \lambda_h. \tag{42}$$

$\square$

## B.2 Derivation of equation 10

In the case of least-square regression, the Legendre-Fenchel conjugate of the least-square loss $f(t; y) = \frac{1}{2}(t - y)^2$ is given by $f^*(s; y) = \frac{s^2}{2} + sy$. Substituting this conjugate function into $H(u)$ in equation 3 in Theorem 2.1, we have
$$H(u) = \max_{v \in \mathbb{R}^n}\left\{-\frac{1}{2\rho}v^\top X D(u) X^\top v - \frac{1}{2}\|v\|_2^2 - v^\top y\right\}. \tag{43}$$
We can verify that the unique optimal solution of $H(u)$ is
$$\hat{v} = -\left(\frac{X D(u) X^\top}{\rho} + I\right)^{-1} y. \tag{44}$$
Substituting $\hat{v}$ back into equation 3 and applying Theorem 2.1 yield the representation
$$L_{\mathrm{BR}} = \min_{(u, z) \in \Omega}\left\{y^\top \left(\frac{1}{\rho}X D(u) X^\top + I\right)^{-1} y\right\}. \tag{45}$$

## B.3 Proof of Corollary 2.3

*Proof of Corollary 2.3.* Under the least-square regression setting, we apply Theorem 2.2 and based on equation 5, we know $\hat{v} = -\left(\frac{X D(\hat{u}) X^\top}{\rho} + I\right)^{-1} y$. We further define $M := \left(I_n + \rho^{-1} X_S X_S^\top\right)$, where $S$ is the support set indicated by $\hat{u}$ and then $\hat{v} = -My$. Substituting $\hat{v}$ by $\hat{v} = -My$ in Theorem 2.2 proves the Corollary. $\square$

## C   PROOF OF THEOREM 2.4

*Proof of Theorem 2.4.* For least-squares loss $f(t, y) = \frac{1}{2}(t - y)^2$, we have that

$$H(u) = \max_{v \in \mathbb{R}^n} \left\{ -\frac{1}{2\rho} v^\top X D(u) X^\top v - \frac{1}{2} \|v\|_2^2 - v^\top y \right\}.$$

Since the optimal value if non-negative, the optimal dual parameter $v \in \mathbb{R}^n$ must satisfy that $\|v\|_2 \leq 2\|y\|_2 \leq 2$.
Note that

$$H(u) - P^* \leq H(u) - H(\bar{u}) = H(u) - H(\tilde{u}) + H(\tilde{u}) - H(\bar{u}), \tag{46}$$

where we set $\tilde{u}_i = \bar{u}_i / \bar{z}_j$ if the feature $i$ belongs to group $j$ and $z_j = 1$, otherwise we set $\tilde{u}_i = 0$.
For $H(u) - H(\tilde{u})$, we have that

$$H(u) - H(\tilde{u})$$

$$= \max_{v \in \mathbb{R}^n} \left\{ -\frac{1}{2\rho} v^\top X D(u) X^\top v - \frac{1}{2} \|v\|_2^2 - v^\top y \right\} - \max_{v \in \mathbb{R}^n} \left\{ -\frac{1}{2\rho} v^\top X D(\tilde{u}) X^\top v - \frac{1}{2} \|v\|_2^2 - v^\top y \right\}$$

$$\leq \max_{v \in \mathbb{R}^n} \left\{ -\frac{1}{2\rho} v^\top X (D(u) - D(\tilde{u})) X^\top v \right\}$$

$$\leq \frac{1}{\rho} \sigma_{\max} \left( X (D(u) - D(\tilde{u})) X^\top \right), \tag{47}$$

where $\sigma_{\max}(\cdot)$ denotes the maximum singular value of the matrix. Similarly, for $H(\tilde{u}) - H(\bar{u})$, we have that

$$H(\tilde{u}) - H(\bar{u}) \leq \frac{1}{\rho} \sigma_{\max} \left( X (D(\tilde{u}) - D(\bar{u})) X^\top \right). \tag{48}$$

For $X(D(\tilde{u}) - D(\bar{u})) X^\top$, we rewrite it as

$$X(D(\tilde{u}) - D(\bar{u})) X^\top = \sum_j \sum_{i \in g_j} \left( z_j \times \frac{\bar{u}_i}{\bar{z}_j} - \bar{u}_i \right) X_i X_i^\top.$$

Note that by our assumption, the operator norm of $\sum_{i \in g_j} \left( z_j \times \frac{\bar{u}_i}{\bar{z}_j} - \bar{u}_i \right) X_i X_i^\top$ is at most $|g_j| \leq \mathcal{G}$ and the mean of the random matrix is $0$. By the Ahlswede-Winter matrix concentration bound Ahlswede & Winter (2002); Oliveira et al. (2010), we have that

$$\Pr \left[ \sigma_{\max} \left( X (D(\tilde{u}) - D(\bar{u})) X^\top \right) \geq \sqrt{r_z} \mathcal{G} t \right] \leq r_z \exp(-\Omega(t^2)), \tag{49}$$

where $r_z$ is the number of fractional entries in $\bar{z}$. The standard matrix concentration bound (e.g., matrix Hoeffding) states that for $n$-dimensional random matrices $X_1, X_2, \ldots, X_M$, we have that $Pr[\sigma_{\max}(X_1 + \cdots + X_M) \geq \alpha] \leq 2n \exp(-\alpha^2/(8M\sigma^2))$, where $\sigma$ upper bounds the operator norms of $X_1, X_2, \ldots, X_M$ almost surely. In the context of Eq. (49), we have that $\sigma = \mathcal{G}$ and we set $\alpha = \sqrt{r_z} \mathcal{G} t$. Using the matrix concentration inequality, we upper bound the Left-Hand-Side of Eq. (49) by $2n \exp(-r_z t^2/8M) \leq 2n \exp(-t^2/8)$ (since $r_z \geq M$). Note that this bound is already good enough since the only difference from the Right-Hand-Side of Eq. (49) is the factor $2n$ instead of $r_z$. These factors would go into the logarithmic factor in the final error bound of Theorem 2.4 so they do not make much difference.
To improve the factor $2n$ to $r_z$ in Eq. (49), we need to show that there exists $r_z$ dimensional subspace whose basis vector denoted by the columns of $Q \in \mathbb{R}^{n \times r_z}$ so that $X(D(\tilde{u}) - D(\bar{u})) X^\top$ can be written as $Q(X_1' + \ldots X_M') Q^\top$ almost surely (where $X_1', \ldots, X_M'$ are $r_z$-dimensional matrices and their operator norms are also bounded by $\mathcal{G}$). The construction of $Q$ and $X_1', \ldots, X_M'$ is possible because each matrix associated with a fractional variable $\bar{z}_j$ is low rank and there are only $r_z$ fractional $\bar{z}_j$'s. We then apply the above matrix concentration bound to $X_1' + \ldots X_M'$ to derive Eq. (49).
For $X(D(u) - D(\tilde{u})) X^\top$, we have that

$$X(D(u) - D(\tilde{u})) X^\top = \sum_i (u_i - \tilde{u}_i) X_i X_i^\top.$$

Again, by our assumption, the operator norm of $(u_i - \tilde{u}_i) X_i X_i^\top$ is at most $1$, and the mean of the random matrix is $0$ (even when conditioned on $z$). Therefore, by the Ahlswede-Winter matrix concentration bound Ahlswede & Winter (2002); Oliveira et al. (2010), we have that

$$\Pr \left[ \sigma_{\max} \left( X (D(u) - D(\tilde{u})) X^\top \right) \geq \sqrt{r_u} t | z \right] \leq r_u \exp(-\Omega(t^2)), \tag{50}$$

where $r_u$ is the number of fractional entries in $\bar{u}$.

Combining equation 46, equation 47, equation 48, equation 49, and equation 50, we have that with probability at least $(1 - \delta)$, it holds that

$$H(u) - P^* \leq O\left(\frac{\sqrt{r_z \log(r_z/\delta)}\mathcal{G} + \sqrt{r_u \log(r_u/\delta)}}{\rho}\right),$$

proving the theorem. $\qquad\square$

# D ANALYSIS OF THE PERFORMANCE OF $L_{\mathrm{BR}}$ ON SYNTHETIC RANDOM ENSEMBLES

## D.1 PROOF OF THEOREM 3.1

*Proof of Theorem 3.1.* Let

$$M = \left(\frac{1}{\rho}XD(u^*)X^\top + I\right)^{-1}.$$

For each feature index $i \in \{1, 2, \ldots, n\}$, by $y = Mw + \epsilon$, we have that

$$e_i^\top X^\top MXy = e_i^\top X^\top MXw + e_i^\top X^\top M\epsilon. \tag{51}$$

*We first bound $e_i^\top X^\top MXw$ as follows.* Applying Lemma D.1 to each feature index $i$ such that $w_i \neq 0$, and via a union bound, we have that there exists a universal constant $c_1 \in (0, 1/3200)$, such that with probability at least $(1 - 3k\exp(-c_1 n^{1-2\delta}))$, it holds that

$$\forall i : w_i \neq 0, \left|\frac{1}{\rho}e_i^\top X^\top MXw\right| \in [0.9/\sqrt{k}, 1.1/\sqrt{k}]. \tag{52}$$

Also, applying Lemma D.2 to each feature index $i$ such that $w_i = 0$, and via a union bound, we have that with probability at least $(1 - 4(d-k)\exp(-c_1 n/k))$, it holds that

$$\forall i : w_i = 0, \left|\frac{1}{\rho}e_i^\top X^\top MXw\right| \leq 0.1/\sqrt{k}. \tag{53}$$

*We then bound $e_i^\top X^\top M\epsilon$ as follows.* Note that $M \preccurlyeq I$. Therefore, applying Lemma D.3 to each feature index $i$ (with $\tau = 0.1\rho/\sqrt{k}$), and via a union bound, we have that with probability at least $(1 - 2d\exp(-n/8) - 2d\exp(-n^{2\delta}/(400\gamma^2 k)))$, it holds that

$$\forall i \in \{1, 2, \ldots, d\}, \left|e_i^\top X^\top M\epsilon\right| \leq 0.1\rho/\sqrt{k}. \tag{54}$$

Now we condition on the event that all of equation 52, equation 53, and equation 54 happen. By equation 51, we have that

$$\forall i : w_i \neq 0 : \left|e_i^\top X^\top MXy\right| \in [0.8\rho/\sqrt{k}, 1.2\rho/\sqrt{k}], \tag{55}$$

$$\forall i : w_i = 0 : \left|e_i^\top X^\top MXy\right| \in [0, 0.2\rho/\sqrt{k}]. \tag{56}$$

We now set $\lambda_k = (0.2\rho/\sqrt{k})^2$, $\lambda_h = ((0.79\rho/\sqrt{k})^2 - \lambda_k) \cdot k_{\min}$ (where we let $k_{\min} = \min_{j \in \{1, 2, \ldots, h\}}\{k_j\}$ be the size of the smallest non-empty group (in terms of non-zero coefficients)), and verify the conditions in Corollary 2.3 as follows:

1. Fix any group $g$ such that $z_g^* = 1$. For each $i \in g$ such that $u_i^* = 1$, we have that $w_i \neq 0$ and therefore $(e_i^\top X^\top MXy)^2 \geq (0.8\rho/\sqrt{k})^2 > \lambda_k$ by equation 55. For each $i \in g$ such that $u_i^* = 0$, we have that $w_i = 0$ and therefore $(e_i^\top X^\top MXy)^2 \leq (0.2\rho/\sqrt{k})^2 \leq \lambda_k$ by equation 56.

2. Fix any group $g$ such that $z_g^* = 1$. By equation 55, we have $\sum_{i \in g:u_i^*=1}((e_i^\top X^\top My)^2 - \lambda_k) \geq ((0.8\rho/\sqrt{k})^2 - \lambda_k) \cdot k_{\min} > \lambda_h$.

3. Fix any group $g$ such that $z_g^* = 0$. By equation 55 and equation 56, we have $\sum_{i \in g} \max\{(e_i^\top X^\top My)^2 - \lambda_k, 0\} \leq 0 < \lambda_h$.

Finally, the theorem is proved by collecting the failure probabilities of the desired events (equation 52, equation 53, and equation 54). $\qquad\square$

## D.2 PROOF OF THEOREM 3.2

*Proof of Theorem 3.2.* Let

$$M = \left(\frac{1}{\rho}XD(u^*)X^\top + I\right)^{-1} = \left(\frac{1}{\rho}X_1X_1^\top + I\right)^{-1} = \left(\frac{1}{\rho}X_2X_2^\top + I\right)^{-1}.$$

For each feature index $i \in \{1, 2, \ldots, n\}$, by $y = Mw^{(1)} + \epsilon$, we have that

$$e_i^\top X^\top MXy = e_i^\top X^\top MXw^{(1)} + e_i^\top X^\top M\epsilon. \tag{57}$$

We first bound $e_i^\top X^\top MXw^{(1)}$ as follows. Applying Lemma D.1 to each feature index $i$ such that $w_i^{(1)} \neq 0$, and via a union bound, we have that there exists a universal constant $c_1' \in (0, 1/3200)$, such that with probability at least $(1 - 3k\exp(-c_1'n^{1-2\delta}))$, it holds that

$$\forall i : w_i^{(1)} \neq 0, \left|\frac{1}{\rho}e_i^\top X^\top MXw^{(1)}\right| \in [0.9|w_i^{(1)}|, 1.1|w_i^{(1)}|]] \subseteq [0.9\xi, 1.1\zeta^{1/4}\xi]. \tag{58}$$

Similarly, applying Lemma D.1 to each feature index $i$ such that $w_i^{(2)} \neq 0$, and via a union bound, we have that with probability at least $(1 - 3k\exp(-c_1'n^{1-2\delta}))$, it holds that

$$\forall i : w_i^{(2)} \neq 0, \left|\frac{1}{\rho}e_i^\top X^\top MXw^{(1)}\right| = \left|\frac{1}{\rho}e_i^\top X^\top MXw^{(2)}\right| \in [0.9|w_i^{(2)}|, 1.1|w_i^{(2)}|]] \subseteq [0.9\xi, 1.1\zeta^{1/4}\xi]. \tag{59}$$

Finally, applying Lemma D.2 to each feature index $i$ such that both $w_i^{(1)} = 0$ and $w_i^{(2)} = 0$, and via a union bound, we have that with probability at least $(1 - 4(d - 2k)\exp(-c_1'n\xi^2))$, it holds that

$$\forall i : w_i^{(1)} = 0 \land w_i^{(2)} = 0, \left|\frac{1}{\rho}e_i^\top X^\top MXw^{(1)}\right| \leq 0.1\xi. \tag{60}$$

We then bound $e_i^\top X^\top M\epsilon$ as follows. Note that $M \preccurlyeq I$. Therefore, applying Lemma D.3 to each feature index $i$ (with $\tau = 0.1\xi\rho$), and via a union bound, we have that with probability at least $(1 - 2d\exp(-n/8) - 2d\exp(-\xi^2n^{2\delta}/(400\gamma^2)))$, it holds that

$$\forall i \in \{1, 2, \ldots, d\}, \left|e_i^\top X^\top M\epsilon\right| \leq 0.1\xi\rho. \tag{61}$$

Now we condition on the event that all of equation 58, equation 59, equation 60, and equation 61 happen. By equation 57, we have that

$$\forall i : w_i^{(1)} \neq 0 \lor w_i^{(2)} \neq 0 : \left|e_i^\top X^\top MXy\right| \in [0.8\xi\rho, 1.2\zeta^{1/4}\xi\rho], \tag{62}$$

$$\forall i : w_i^{(1)} = 0 \land w_i^{(2)} = 0 : \left|e_i^\top X^\top MXy\right| \in [0, 0.2\xi\rho]. \tag{63}$$

We now set $\lambda_k = (0.2\xi\rho)^2$, $\lambda_h = ((0.79\xi\rho)^2 - \lambda_k) \cdot (k/h)$, and verify the conditions in Corollary 2.3 as follows:

1. Fix any group $g$ such that $z_g^* = 1$. For each $i \in g$ such that $u_i^* = 1$, we have that $w_i^{(1)} \neq 0$ and therefore $(e_i^\top X^\top MXy)^2 \geq (0.8\xi\rho)^2 > \lambda_k$ by equation 62. For each $i \in g$ such that $u_i^* = 0$, we have that $w_i^{(1)} = w_i^{(2)} = 0$ and therefore $(e_i^\top X^\top MXy)^2 \leq (0.2\xi\rho)^2 \leq \lambda_k$ by equation 63.

2. Fix any group $g$ such that $z_g^* = 1$. By equation 62, we have $\sum_{i \in g:u_i^*=1}((e_i^\top X^\top My)^2 - \lambda_k) = ((0.8\xi\rho)^2 - \lambda_k) \cdot (k/h) > \lambda_h$.

3. Fix any group $g$ such that $z_g^* = 0$. By equation 62 and equation 63, we have $\sum_{i \in g} \max\{(e_i^\top X^\top My)^2 - \lambda_k, 0\} \leq ((1.44\xi\rho)^2\sqrt{\zeta} - \lambda_k) \cdot (k/(\zeta h)) \leq ((1.44\xi\rho)^2 - \lambda_k) \cdot (k/(\sqrt{\zeta}h)) = 2.0336(\xi\rho)^2 \cdot (k/h)/\sqrt{\zeta} < \lambda_h$, where the last inequality holds for large enough constant $\zeta > 1$.

Finally, the theorem is proved by collecting the failure probabilities of the desired events (equation 58, equation 59, equation 60, and equation 61). $\square$

## D.3 TECHNICAL LEMMAS

**Lemma D.1.** *Suppose $k \leq n/4$. Let $X \in \mathbb{R}^{n \times k}$ be a matrix with i.i.d. $\mathcal{N}(0, 1)$ entries. Let $\rho = n^{1/2+\delta}$ ($\delta \geq 0$), and $M = (I + \frac{1}{\rho}XX^\top)^{-1}$. For any $\epsilon \geq 32n^{\delta-1/2}$, any fixed vector $z \in \mathbb{R}^k$ such that $\|z\|_\infty \leq 1$ and any fixed index $i \in \{1, 2, \ldots, k\}$, with probability at least*

$(1 - 3\exp\left(-n^{1-2\delta}\epsilon^2/2048\right))$, *it holds that*

$$\left|e_i^\top \left(\frac{1}{\rho}X^\top MX - I\right) z\right| \leq \epsilon,$$

*where $e_i$ is the $i$-th (column) basis vector.*

*Proof.* The proof follows the similar lines of Part (1) of the proof of Lemma 2 in Pilanci et al. (2015). However, we adopt a different regularization parameter $\rho$.

We write $X = UDV^\top$ for the singular decomposition of $X$. By standard results on the singular value of Gaussian random matrices (e.g., Davidson & Szarek (2001)), for each $t \geq 0$, it holds that

$$\Pr[\forall j \in \{1,2,\ldots,k\} : \sqrt{n} - \sqrt{k} - t \leq D_{jj} \leq \sqrt{n} + \sqrt{k} + t] \geq 1 - 2\exp(-t^2/2).$$

In particular, if we set $t = \sqrt{n}/4$, we have that

$$\Pr[\forall j \in \{1,2,\ldots,k\} : \sqrt{n}/4 \leq D_{jj} \leq 7\sqrt{n}/4] \geq 1 - 2\exp(-n/32). \tag{64}$$

The rest of the proof will be carried out by conditioning on the successful event in equation 64. Note that

$$X^\top MX = V(\rho I + D^2)^{-1} D^2 V^\top,$$

and therefore,

$$\frac{1}{\rho}X^\top MX - I = V[(\rho I + D^2)^{-1}D^2 - I]V^\top = V\tilde{D}V^\top,$$

where we let $\tilde{D} := \mathrm{diag}(\{\frac{D_{jj}^2}{\rho+D_{jj}^2} - 1\}_{j\in\{1,2,\ldots,k\}})$, and have that $\tilde{D}_{jj} \geq 0$ and $\tilde{D}_{jj} \leq 16n^{\delta-1/2}$ for all $j \in \{1,2,\ldots,k\}$ due to the successful event in equation 64. Note that

$$\left|e_i^\top \left(\frac{1}{\rho}X^\top MX - I\right) z\right| = \left|e_i^\top V\tilde{D}V^\top z\right| = \left|\sum_j V_{ij}\tilde{D}_{jj}\sum_q V_{qj}z_q\right|$$

$$\leq \left|\sum_j V_{ij}^2 \tilde{D}_{jj}z_i\right| + \left|\sum_j V_{ij}\tilde{D}_{jj}\sum_{q:q\neq i} V_{qj}z_q\right|. \tag{65}$$

It is easy to bound the first term in equation 65 by

$$\left|\sum_j V_{ij}^2 \tilde{D}_{jj}z_i\right| \leq 16n^{\delta-1/2}. \tag{66}$$

Let $\tilde{V}$ be the $(k-1) \times k$ matrix obtained by removing the $i$-th row from $V$, and let $\tilde{z}$ be the $(k-1)$-dimensional vector by removing the $i$-th entry of $z$. We can rewrite the second term in equation 65 as

$$\left|\sum_j V_{ij}\tilde{D}_{jj}\sum_{q:q\neq i} V_{qj}z_q\right| = \left|e_i^\top V\tilde{D}\tilde{V}^\top \tilde{z}\right|. \tag{67}$$

Observe that even when conditioned on $D$ (and therefore $\tilde{D}$), $e_i^\top V\tilde{D}\tilde{V}^\top$ is a $(k-1)$-dimensional vector pointing towards a uniform random direction, and its 2-norm is at most $16n^{\delta-1/2}$. On the other hand, $\tilde{z}$ is a fixed $(k-1)$-dimensional vector with $\|\tilde{z}\|_2 \leq \sqrt{k-1}$. Therefore, by standard spherical concentration inequality, we have that

$$\Pr\left[\left|e_i^\top V\tilde{D}\tilde{V}^\top \tilde{z}\right| \leq \epsilon - 16n^{\delta-1/2}\right] \geq 1 - \exp\left(-n(\epsilon - 16n^{\delta-1/2})^2/512\right)$$

$$\geq 1 - \exp\left(-n^{1-2\delta}\epsilon^2/2048\right). \tag{68}$$

Combining equation 65, equation 66, equation 67, equation 68, and collecting the probabilities, we prove the desired result. □

**Lemma D.2.** *Suppose $k \leq n/4$. Let $X \in \mathbb{R}^{n\times k}$ be a matrix with i.i.d. $\mathcal{N}(0,1)$ entries. Let $u \in \mathbb{R}^k$ be a column vector with i.i.d. $\mathcal{N}(0,1)$ entries. Let $\rho = n^{1/2+\delta}$ ($\delta \geq 0$), and $M = (I + \frac{1}{\rho}XX^\top)^{-1}$. For any $\epsilon \in (0,1)$ and any fixed vector $z \in \mathbb{R}^k$ such that $\|z\|_2 \leq 1$, with probability at least $(1 - 4\exp\left(-n\epsilon^2/32\right))$, it holds that*

$$\left|\frac{1}{\rho}u^\top MXz\right| \leq \epsilon.$$

*Proof.* The following proof is based on the standard calculation, which also appeared in Part (2) of the proof of Lemma 2 in Pilanci et al. (2015).

Write $X = UDV^\top$ for the singular decomposition of $X$. Again, we have equation 64 and will condition on the successful event in equation 64 for the rest of the proof. Note that

$$\frac{1}{\rho} MXz = U(\rho D^{-1} + D)^{-1} V^\top z,$$

and therefore

$$\left\| \frac{1}{\rho} MXz \right\|_2 \le 4/\sqrt{n}$$

by the successful event in equation 64. Thus, $\frac{1}{\rho} u^\top MXz$ is a centered Gaussian with standard deviation $4/\sqrt{n}$, and the probability that $\left| \frac{1}{\rho} u^\top MXz \right| > \epsilon$ is at most $2\exp(-\epsilon^2 n/32)$ by the standard Gaussian tail bound. The lemma is proved by collecting the failure probabilities. □

**Lemma D.3.** *Let $u \in \mathbb{R}^n$ be a column vector with i.i.d. $\mathcal{N}(0,1)$ entries. Let $M \in \mathbb{R}^{n \times n}$ be any PSD matrix (which might depend on $u$) such that its eigenvalues are at most 1. Let $\epsilon \in \mathbb{R}^n$ be an independent column noise vector with i.i.d. $\mathcal{N}(0, \gamma^2)$ entries. For any $\tau > 0$, with probability at least $(1 - 2\exp(-n/8) - 2\exp(-\tau^2/(4\gamma^2 n)))$, it holds that*

$$\left| u^\top M\epsilon \right| \le \tau.$$

*Proof.* By standard $\chi^2$ concentration results, we have that with probability at least $(1 - 2\exp(-n/8))$, it holds that $\|u\|_2^2 \le 2n$. The rest of the proof will be carried out conditioning on this event. Note that $\|Mu\|_2 \le \|u\|_2$. Therefore, $u^\top M\epsilon \sim \mathcal{N}(0, \gamma^2 \|u\|_2^2)$. By the standard Gaussian tail bound, we have that

$$\Pr[|u^\top M\epsilon| \le \tau] \ge 1 - 2\exp(-\tau^2/(2\gamma^2 \|u\|_2^2)) \ge 1 - 2\exp(-\tau^2/(4\gamma^2 n)).$$

The lemma is proved by collecting the failure probabilities. □

# E  OPTIMIZATION

We use the projected Quasi-Newton (PQN) method to solve the optimization $L_{\mathrm{BR}}$ defined in equation 10. The details of PQN is elaborated in Schmidt et al. (2009) Algorithm 1, therefore, we refer the interested audiences to Schmidt et al. (2009) for more details. To apply PQN, we need to know the gradient of the objective function in equation 10. The partial gradient of $G(u)$ w.r.t $u_i$ can be written as

$$\frac{\partial G(u)}{\partial u_i} = -\frac{1}{\rho} \left( X_i^\top \left( \frac{1}{\rho} XD(u)X^\top + I \right)^{-1} y \right). \tag{69}$$

Computing such a gradient requires the solution of a rank-$\|u\|_0$ linear system of size $n$, which can be calculated in time $\mathcal{O}(\|u\|_0^3) + \mathcal{O}(nd)$ via the QR decomposition. When the sparsity level $k$ is relatively small, such computation is not expensive.

We also need to do the following projection in PQN.

$$\min_{x \in \Omega} : \|x - y\|_2^2, \tag{70}$$

where $\Omega$ is defined in Section 2.2. The projection on the relaxed constraint set $\Omega$ can be efficiently obtained by a commercial solver (we use Gurobi Gurobi Optimization, LLC (2022)).

# F  NUMERICAL VERIFICATION OF COROLLARY 2.3

Corollary 2.3 states that under the least-squares regression setting, under certain conditions, the solution of the relaxed program is integral and consistent with the optimal solution of the problem with Boolean constraints. In this section, we numerically verify the equivalence established in Corollary 2.3.

We consider to generate synthetic data follow Random Ensemble I. we set $d = 200$, $b = 10$, $k = 50$ ($k_i = 10, \forall i \in \{1, 2, \dots, 5\}$), $h = 5$. We vary sample size $n$ and SNR (controlled by $\gamma$) to see whether the solution of the relaxed program is the optimal integral solution of the program with Boolean constraints.

For each $n$ and SNR, we randomly generate 10 datasets. We run our relaxed program on these 10 datasets and count the number of solutions that are the same as the integral solutions of the program with Boolean constraints. As shown in the Table S1, the proposed equivalence can be achieved when the sample size and SNR are large.

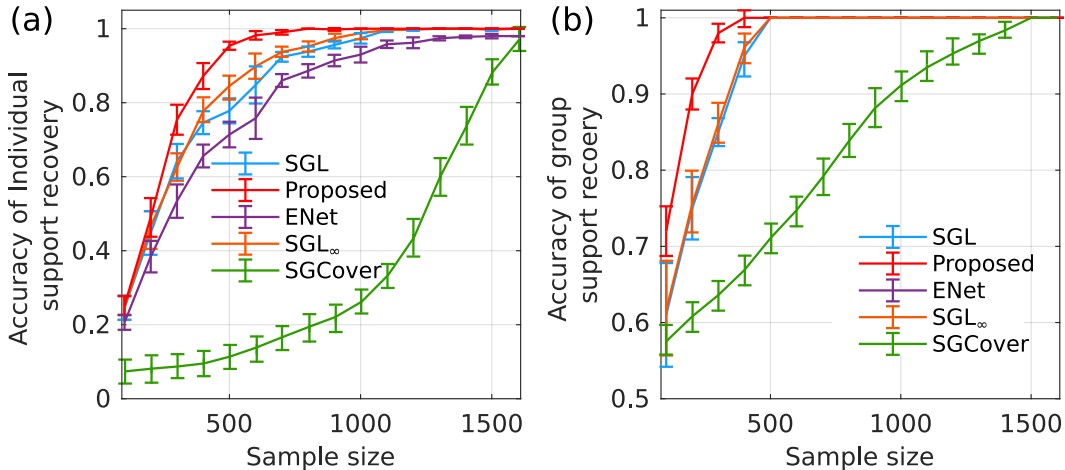

Figure S1: Performance comparison for different $\gamma$s. (a) $A_I$ as $n$ increases when $\gamma = 3.5$. (b) $A_G$ as $n$ increases when $\gamma = 3.5$. We average results over 10 datasets and the error bar means 95% confidence interval.

Table S1: Number of solutions that is the integral solutions of the program with Boolean constraints .

| SNR | $n = 100$ | $n = 200$ | $n = 300$ | $n = 1,000$ |
|-----|-----------|-----------|-----------|-------------|
| 40  | 0/10      | 10/10     | 10/10     | 10/10       |
| 30  | 0/10      | 10/10     | 10/10     | 10/10       |
| 20  | 0/10      | 0/10      | 0/10      | 10/10       |

## G  ADDITIONAL EXPERIMENTS FOR RANDOM ENSEMBLE I

We compare our proposed method with other state-of-the-art methods on simulation data generated from Random Ensemble I with different $\gamma$s. We first set $\gamma = 3.5$ and compare the performance of the competing methods on recovering the individual and group supports. Note that when $\gamma = 3.5$, the signal to noise ratio is around 3. We compare the competing methods on 10 different data sets generated from Random Ensemble I with $\gamma = 3.5$. As illustrated in Fig. S1 (a) and (b), our proposed method still outperforms all the competing methods in terms of both $A_I$ and $A_G$ with the increasing number of samples.

## H  EXPERIMENTS FOR SYNTHETIC DATA SATISFYING MUTUAL INCOHERENCE CONDITION

In Random Ensemble I, we generate $x_i \sim \mathcal{N}(0, I)$ as i.i.d features. In this experiments, we aims to evaluate the performance of the competing methods in the presence of correlation between features. We follow the way we generate the data in Random Ensemble I, where we set $d = 1000$, $b = 10$, $k = 50$ ($k_i = 10, \forall i \in \{1, 2, \ldots, 5\}$), $h = 5$, and $\gamma = 0.1$. Then only different is that we generate $x_i \sim \mathcal{N}(0, \Sigma)$, where $\Sigma$ is the Toeplitz covariance matrix $\Sigma_{ij} = p^{|i-j|}$. Such matrices satisfy the mutual incoherence condition, required by $\ell_1$-regularized estimators to be statistically consistent. We consider $p = 0.2$ and $p = 0.7$ for $\Sigma$ to evaluate the competing methods' performance under different correlations. The performance is shown in Fig. S2. We find that when $p = 0.2$, support recovery (both individual level Fig. S2(a) and group level Fig. S2(b)) can be easily achieved by most of the models. And our model outperforms other competing methods. However, when $p = 0.7$, all models have difficulties to accurately recover the support at the individual level (Fig. S2(c)), which makes sense because the correlation between features are high so that finding the correct features becomes more challenge. Our model still outperforms other competing methods at both individual level (Fig. S2(c)) and group level (Fig. S2(d)).

## I  ADDITIONAL EXPERIMENTS FOR CANCER DRUG RESPONSE PREDICTION

We first show Table. S3, which illustrates the pathways and genes identified by the proposed method for drug IMATNIB and the corresponding researches that support the findings.
We further find the targeted pathways and genes for three other drugs: GEFITINIB, BEXAROTENE, and BOSUTINIB. In Table. S4, we show the number of targeted pathways and genes identified by

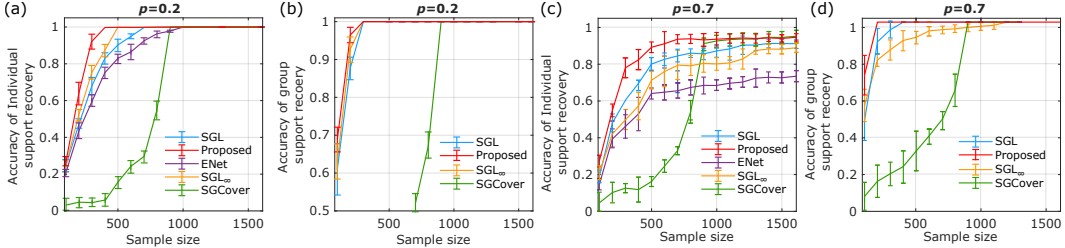

Figure S2: Performance comparison for different $p$s. (a) $A_I$ as $n$ increases when $p = 0.2$. (b) $A_G$ as $n$ increases when $p = 0.2$. (c) $A_I$ as $n$ increases when $p = 0.7$. (d) $A_G$ as $n$ increases when $p = 0.y$. We average results over 10 datasets and the error bar means 95% confidence interval for (a)-(d).

each competing method and the out-of-sample MSE. As shown our method achieves the smallest out-of-sample MSE and the fewest number of pathways and genes.

## J   TIME COMPARISON FOR RANDOM ENSEMBLE I

We compared the running time for experiments of Random Ensemble I when the sample size is 1,000 in the Table. S2.

Table S2: Time comparison (mean $\pm$ s.d.) for Random Ensemble I when the sample size is 1000.

| Method | SGL | SGL_$\infty$ | SGCover | ENEt | Proposed |
|---|---|---|---|---|---|
| Time (s) | 1.4$\pm$0.1 | 4.2$\pm$ 0.2 | 20.8$\pm$0.9 | 0.2$\pm$0.03 | 24.4$\pm$2.1 |

## K   IMPLEMENTATION DETAILS

For section 4.1 subsection "Feature selection with given support sizes $k$ and $h$" and section 4.2, we select the parameters as follows. For our method, because $k$ and $h$ are given, we only have one parameter $\rho$ in (1) left, we select $\rho$ by the 5-fold CV in terms of MSE. For the result of the methods, which cannot control $k$ and $h$, we just sweep the parameters to let them yield the desired $k$ and $h$. For the real-world application, we select the parameters in terms of out-of-sample MSE.

## L   CODE AVAILABILITY

The codes for the proposed method can be found here: `https://anonymous.4open. science/r/L0GL-F107/Readme`

Table S3: Pathways and genes identified by the proposed methods for IMATNIB.

| Pathway | Genes | Reference |
|---|---|---|
| RHO GTPases Activate WASPs and WAVEs | ARPC1B WASF1 ARPC5 WASL CYFIP1 ACTG1 ACTR3 | Gu et al. (2009); Huang et al. (2008); Chen et a |
| Regulation of PTEN gene transcription | LAMTOR3 LAMTOR4 SNAI1 RPTOR RRAGA RRAGB MBD3 RRAGD PHC3 GATAD2A RCOR1 MECOM CBX8 LAMTOR2 | Nishioka et al. (2011); Peng et al. (2010); Huan |
| Signaling by PDGF | PDGFC COL4A3 COL6A2 COL6A3 COL9A3 | Malavaki et al. (2013); Li et al. (2006); Heldin |
| Retinoid metabolism and transport | CLPS LRP8 APOC3 SDC4 LPL LRP10 LRP12 APOA2 | Hoang et al. (2010) |
| TCF transactivating complex | RBBP5 KAT5 PYGO1 PYGO2 BCL9 | Zhang et al. (2021); Coluccia et al. (2007); Co |
| Deactivation of the beta-catenin transactivating complex | RBBP5 SOX3 SRY PYGO1 PYGO2 CBY1 BCL9 | Zhou et al. (2003); Leo et al. (2013) |
| RAS processing | ZDHHC9 GOLGA7 BCL2L1 ABHD17B | Chung et al. (2006); Braun & Shannon (2008) |

Table S4: Result comparison for three other drugs.

| Durg | Method | $k$ (s.d.) | $h$ (s.d.) | Out-of-sample MSE $\pm$ 95% CI |
|---|---|---|---|---|
| BOSUTINIB | Proposed method | 36 | 5 | 29.4$\pm$2.1 |
| | SGL-Overlap | 64 ( 4.3) | 9 (0.9) | 42.6 $\pm$ 2.3 |
| | ENet | 48 (5.3) | 18(3.2) | 35.4 $\pm$ 3.1 |
| | SGCover | 240 (10.4) | 15 (1.8) | 52.4 $\pm$ 4.2 |
| GEFITINIB | Proposed method | 42 | 6 | 35.4 $\pm$ 1.4 |
| | SGL-Overlap | 78 (4.6) | 13 (1.3) | 4.78 $\pm$ 2.1 |
| | ENet | 49 (5.6) | 15 (2.1) | 38.7 $\pm$ 2.9 |
| | SGCover | 278 (13.4) | 12 (2.4) | 59.6 $\pm$ 3.6 |
| BEXAROTENE | Proposed method | 52 | 8 | 36.9 $\pm$ 1.8 |
| | SGL-Overlap | 86 (3.8) | 15 (1.5) | 61.8 $\pm$ 2.7 |
| | ENet | 64 (5.2) | 17 (3.5) | 38.4 $\pm$ 2.3 |
| | SGCover | 312 (16.3) | 18 (2.3) | 58.2 $\pm$ 3.1 |

