# OpenReview forum: "Learning Sparse Group Models Through Boolean Relaxation"
_ICLR.cc/2023/Conference — ICLR 2023 notable top 25%_

### Official Review · Reviewer_fvhG · 2022-10-23

**Confidence:** 3
**Correctness:** 3
**Technical Novelty And Significance:** 3
**Empirical Novelty And Significance:** 3
**Recommendation:** 6

**Clarity, Quality, Novelty And Reproducibility:**

The paper is clearly written, except that the motivation for the method could be done in more detail. The formulation is new -- builds up on prior work, but the extension is significant.  Experiments are described well, and code is provided -- so seems reproducible. Practical significance hinges on computational complexity -- which isn't discussed / addressed in the paper.

**Strength And Weaknesses:**

Strength:  The so-called boolean relaxation is qualitatively different from the large existing body of work on convex relaxations of individual and group sparsity.  The paper extends this approach from regular sparsity, done in prior work, to group sparsity.  Very thorough theoretical analysis, and good experimental performance on simulated and real data.  Also a so-called 'equivalent' condition for non-overlapping groups is proposed which certifies that the convex relaxation is equivalent to the binary optimization problem -- although I'm not sure if it's realistic to evaluate it in practice, or it has purely theoretical value. Overall, group-sparse priors are important in practice, and if the method is indeed significantly better in terms of support and group-support recovery in the low-data regime, that could be impactful.

Weaknesses:
1.  Limited or no discussion of computational complexity + timing results.  How does the method compare to enet / Sparse-group-lasso and related methods.  From the construction it appears significantly more expensive.  Even if it's more expensive -- it could still be valuable in settings with limited data -- please add some discussion / timing results.

2. The explanation of the main method itself in the main body of the paper is rather concise, with limited motivation.  Essentially the formulation is just given onto the reader.  Due to the very tight reviewing deadline -- I haven't had a chance to read the supplementary material -- if there's additional explanatory detail there, it would be helpful to move it upfront.

3. Is the result in theorem 2.2. -- equivalent condition -- possible to evaluate numerically for a given problem? How would you find the lambda's?  Also you claim there are no equivalent recovery results ("rigorous theoretical techniques" ) for group-sparse settings (GL and the like) -- what do you mean exactly -- there's a large literature on this topic, with various group-generalizations of conditions from plain sparsity for exact recovery.

4.  You mentioned that the formulation captures other structural sparsity assumptions -- like tree-sparsity, DAGs, e.t.c.  Does this possibly require a very large number of groups in your formulation (so only of theoretical interest)?

5. Random ensemble II is curious -- but highly artificial.

6. How well can you control k and h by sweeping regularization parameters for competing methods?  What do you mean by ranking results (based on coefficient magnitude) for other methods?  The ability to pick exact k and h is a nice advantage -- but I wonder if the improvement in recovery results is somehow an artifact of not being able to find parameters to get the right solution for GL-family, and instead of forcing an incorrect solution to have the same k and h?

7. Have you looked at greedy (of the group-OMP type) or  iterative-hard-thresholding methods for group-sparsity?  These can control exact group sparsity, and some do come with theoretical analysis.  How do they compare with the proposal?



**Summary Of The Paper:**

The paper proposes a new convex relaxation for the high-dimensional group-sparse recovery problem.  It builds on the boolean relaxation from element-wise sparsity from previous work.  In experiments the method is shown to improve individual feature and group-recovery upon existing sparse-group-lasso family of methods in the low-data regime.  It also allows direct control of the desired (group)-sparsity instead of more cumbersome sweep over regularization parameters.


**Summary Of The Review:**

Interesting work on group-sparse recovery, a formulation that stands out from the existing (large) body of work.  It naturally builds on the boolean relaxation for individual sparsity done in prior work, but the extension is not trivial (especially the analysis -- which I unfortunately couldn't check carefully due to the extremely short timeline of the review process).  Empirically the proposed method appears to give noticeably improved recovery performance in low-sample regime.  I am unclear of the practical significance -- as computational complexity / timing results are not provided, and likely high.

---

### Official Review · Reviewer_4Ewe · 2022-10-25

**Confidence:** 2
**Correctness:** 3
**Technical Novelty And Significance:** 3
**Empirical Novelty And Significance:** 3
**Recommendation:** 8

**Clarity, Quality, Novelty And Reproducibility:**

- Clarity: I found the paper to be well-written and easy to follow.
- Novelty: the propose algorithm is novel and extends the related work Pilanci et al. (2015).
- Quality: theoretical justifications are solid.

**Strength And Weaknesses:**

The idea of the paper is clear and well-presented. The proposed Boolean relaxation supports a variety of models with proper Legendre-Fenchel conjugates. The authors also illustrate the utility of the method in least-squares regression. Empirically, the authors demonstrated that the algorithm could help recover the support sizes k and h under certain circumstances. It could be useful to provide the computational complexity of the algorithm. At a first glance, the complexity appears to be O(n^2 d^2) which could be relatively expensive.


**Summary Of The Paper:**

The paper presents an algorithm for learning sparse group models. The key technique used is a convex Boolean relaxation. Solution to the resulting Boolean relaxation is fractional, and the authors then propose rounding algorithms to recover the feasible Boolean solutions with provable guarantees.

**Summary Of The Review:**

I enjoyed reading the paper, and recommend acceptance.

---

### Official Review · Reviewer_7cXz · 2022-10-26

**Confidence:** 3
**Correctness:** 3
**Technical Novelty And Significance:** 2
**Empirical Novelty And Significance:** 3
**Recommendation:** 8

**Clarity, Quality, Novelty And Reproducibility:**

The paper is well-organized and reads well. The novelty is fine. The proposed method is an extension of the Boolean relaxation method from general sparsity to structured sparsity.

**Strength And Weaknesses:**

Strengths:
1. The paper provides theoretical guarantees on the equivalence between the original problem and the relaxed problem, which makes the relaxation more convincing.

2. The paper provides two examples to demonstrate equivalence.

Weakness:
1. Theorems 3.1 and 3.2 claim that the relaxed program achieves the optimal solution. However, such a solution is only consistent with the underlying regression vector $w$. It is unclear whether this solution is consistent with the optimal solution of the problem with Boolean constraints, i.e., Problem (3). If not, the equivalence is not established actually.

2. Experimental verification for the established equivalence is necessary. More specifically, the author should design experiments to show that the solution of the relaxed program is integral and consistent with the optimal solution of the problem with Boolean constraints.

3. In Theorem 3.1, $\rho$ should be $n^{1/2 + \delta}$ instead of $n^{1/2} + \delta$.

4. The paper presents a rounding scheme. It is unclear whether the rounding scheme has been applied in experiments.






**Summary Of The Paper:**

This paper considers the problem of learning sparse group models via Boolean convex relaxation. The paper establishes the equivalence between the original problem with the cardinality constraint and the relaxed convex problem with the Boolean relaxation. Then, the authors apply the established equivalence to two ensembles of random problem instances and prove that the proposed relaxation method can achieve the true support of the regression vector.

**Summary Of The Review:**

Overall the paper is interesting, which studied the Boolean relaxation for sparse group models and established the theoretical equivalence between the original problem and the relaxed problem. However, the authors should solve the concerns about Theorems 3.1 and 3.2, and provide experimental verification (Please refer to the Weakness part for more details).

---

### Official Review · Reviewer_4rdY · 2022-10-27

**Confidence:** 3
**Correctness:** 4
**Technical Novelty And Significance:** 3
**Empirical Novelty And Significance:** 2
**Recommendation:** 8

**Clarity, Quality, Novelty And Reproducibility:**

Clarity - The paper is easy to follow

Quality / Novelty - this is an interesting take on learning sparse group structured models. May be worth investigating the similarities to the references quoted earlier in this review.




**Strength And Weaknesses:**

Strengths
1. The paper is written well and easy to follow.
2. Presents theoretical results on support recovery and convergence

Questions / Comments
1. The idea has some similarity to (I) "Convex relaxation for combinatorial penalties - Obozinski. et. al, 2012" and (ii)"Identifying Groups of Strongly Correlated Variables through Smoothed Ordered Weighted 𝐿1-norms - Sankaran et.al 2017". The above works consider L_p relaxations of combinatorial (submodular) penalties, the special case of which is L_2 relaxation of the penalties. The formulation proposed in this submitted paper may be compared to (i) and (ii) after converting the constraints into a penalty.

2. Most/All the results are presented for non-overlapping groups only, even though the problem setup initially studied is in general. It may be better to stress this in the paper to avoid misrepresentation.

3. The random ensembles are studied for iid design matrices. May be interesting to understand how the algorithms compare in the presence of correlation within columns of X.

4. What are the regimes where the other compared algorithms such as SGL, SGL_\infty may do better in the structure recovery ? Or, is the proposed formulation expected to perform better than them in many other settings than the ones considered in this paper. If so, how do we reason out on the performance ?


**Summary Of The Paper:**

In this paper, the authors propose learning group structured sparsity using explicit constraints on the feature and group sparsity. They derive a reformulation of the proposed problem in terms of boolean variables for the selected features and groups, which is then converted into a convex problem by relaxing the boolean constraints to lie within [0, 1]. Theoretical results are presented to (a) give conditions when the relaxed problem recovers the true model structure, (b) special cases when the model and design matrix are generated according to a random ensemble protocol.
The experimental results illustrate the superiority of the proposed scheme against similar methods such as SGL, SGL_\infty.

**Summary Of The Review:**

Overall, this paper presents an approch for learning group structured sparse models. The theoretical results and evaluation are convincing.

---

### Decision · Program_Chairs · 2023-01-20

**Decision:**

Accept: notable-top-25%

**Justification For Why Not Higher Score:**

* The problem setting may be a bit "niche" for the overall community.

**Justification For Why Not Lower Score:**

* Submission unanimously supported by the reviewers.
* Original formulation with solid theoretical justification.
* Manuscript of excellent quality (writing, structure, clarity).

**Metareview: Summary, Strengths And Weaknesses:**

The reviewers and meta reviewer all appreciated the quality of the work with a clear, well-written manuscript and an original formulation of group-sparse recovery in the form of a boolean relaxation. Moreover, the method is supported by a solid theoretical justification.
They thank the authors for their response and their efforts during the rebuttal phase, which further improves the submission (e.g., discussion about the complexity). The reviewers and meta reviewer unanimously recommend the paper for acceptance.


**Note From Pc:**

if the above contains the word "oral" or "spotlight" please see: "oral" presentation means -> notable-top-5% and "spotlight" means -> notable-top-25%. As stated in our emails, we are disassociating presentation type from AC recommendations